# Regularity and Stability Properties of Selective SSMs with Discontinuous Gating

**Nikola Zubić**                                                        *zubic@ifi.uzh.ch*
*Robotics and Perception Group*
*University of Zurich*

**Davide Scaramuzza**                                                   *sdavide@ifi.uzh.ch*
*Robotics and Perception Group*
*University of Zurich*

Reviewed on OpenReview: *https://openreview.net/forum?id=7Vav53cDeN*

## Abstract

Selective State-Space Models (SSMs) such as Mamba have become central to long-sequence modeling. Still, their stability is poorly understood: their state-space coefficients are modulated online by a token-dependent gating signal, making the recurrence neither linear time-invariant nor classically nonlinear. We study continuous-time selective SSMs through passivity, dissipativity, and Input-to-State Stability (ISS), explicitly separating the *selection signal* $x(\cdot)$ from the *driving input* $u(\cdot)$. We obtain four results: exponential forgetting under strict dissipativity; a canonical $\mathrm{AUC}_{\mathrm{loc}}$ quadratic storage for the frozen-selection subsystem that accommodates discontinuous gating; a parametric LMI together with universal kernel constraints and "irreversible forgetting" under universal quadratic storage; and sufficient conditions for global ISS uniformly over admissible selection schedules. We then bridge to practice by deriving a sampled block LMI for the Mamba selective-scan core, which is used as a differentiable training-time regularizer. Across seven standard time-series datasets and four prediction horizons, the regularizer reduces sampled Mamba-core LMI violations by roughly 92% in 28/28 pairs at a clean-MSE cost of less than 0.018%. It improves internal Mamba passivity and state-norm diagnostics under injected perturbations. Our results turn classical control-theoretic tools into verifiable structural and training criteria for selective SSMs, while honestly scoping which guarantees transfer to a deep selective-scan architecture.

## 1 Introduction

Modern long-sequence models such as Mamba (Gu & Dao, 2024), S7 (Soydan et al., 2024), HGRN (Qin et al., 2024), and GLA (Yang et al., 2024) are usually described as state-space recurrences whose coefficients are not fixed but are computed online from each token. Concretely, in a continuous-time abstraction of these architectures, we will work with

$$\begin{cases} \dot{h}(t) = A\big(\Delta(t), x(t)\big) h(t) + B\big(\Delta(t), x(t)\big) u(t), \\ y(t) = C\big(\Delta(t), x(t)\big) h(t), \end{cases} \tag{1}$$

where $h(t) \in \mathbb{C}^N$ is the recurrent state, $\Delta(t)$ is a gating signal, $x(t)$ is a *selection* (or scheduling) signal that determines the state-space matrices, and $u(t)$ is the driving (port) input that appears in the energy supply rate. In token-selective architectures, it is natural to set $x(\cdot) \equiv u(\cdot)$ (token, gate, and drive collapse into a single signal), but, as we will see, it is analytically much cleaner to keep them separate.

The selectivity of $A, B, C$ on $x(t)$, and the typically piecewise-constant nature of the gating $\Delta(t)$, place equation 1 in an awkward position. It is not a linear time-invariant system, nor a classical linear time-varying (LTV) system, because the time-varying coefficients are themselves data-dependent, and it is not a

classical smooth nonlinear control system either, because gating introduces *discontinuities*. Existing analyses of these architectures focus mostly on expressivity (Cirone et al., 2024; Zubic et al., 2025; Zubić et al., 2026) or on Lyapunov-exponent-style stability arguments that treat the recurrence as a fixed dynamical system per input (Halloran et al., 2024). What is missing is a unified *energy-based* account: when do these models forget initial conditions, how should we measure "how much of the state matters", and what does any of this tell us about how to design or regularize a real Mamba implementation?

Passivity and dissipativity, in the sense of Willems (Willems, 1972; Van der Schaft, 2000), and Input-to-State Stability (ISS) (Sontag et al., 1989; Sontag & Wang, 1995; Jiang et al., 1999), were developed exactly for systems whose state interacts with an external port via a supply rate $\Re\langle u, y\rangle$. They are robust to time variation, they have a natural notion of energy storage, and recent work (Morandin & Hinsen, 2024) has shown that for passive LTV systems the minimal available storage is automatically quadratic, with controlled regularity, even when the coefficients are merely $L^p_{\mathrm{loc}}$. This is the right setting for selective SSMs: gating produces $L^p_{\mathrm{loc}}$ (in fact piecewise-constant) coefficients, and we want to talk about energy stored in the recurrent state $h(t)$.

A key conceptual move in this paper is to keep two signals separate. The selection signal $x(\cdot)$ *schedules* the dynamics through $A, B, C$, while the port input $u(\cdot)$ *drives* the dynamics and appears in the supply rate $\Re\langle u, y\rangle$. In a token-selective architecture, both roles are played by the same vector, $x \equiv u$, but treating them as one signal hides the structure of the problem: passivity and ISS quantifiers are statements about *all admissible u*, while structural results about gating are statements about *all admissible x at almost every time*. Once these quantifiers are separated, the LMI we will derive admits a natural reading as a parametric condition that must hold for every selection value $x$ that gating can produce, while $u$ is absorbed by quadratic completion.

| Standard ingredients (used) | What this paper adds (new) |
|---|---|
| Dissipative systems and passivity (Willems, 1972; Van der Schaft, 2000). | A clean separation of selection $x(\cdot)$ vs. port input $u(\cdot)$ for selective SSMs, and a derivation of the implied passivity constraints. |
| ISS theory and ISS-Lyapunov functions (Sontag et al., 1989; Jiang et al., 1999). | Global ISS w.r.t. $u(\cdot)$ uniformly over admissible selection schedules, under a uniform LMI on $(A, B, C)$ in $x$. |
| Quadratic storage for passive LTV systems with $L^p_{\mathrm{loc}}$ coefficients (Morandin & Hinsen, 2024). | Specialisation to the *frozen-selection* subsystem $x(t) \equiv 0$ of a selective SSM, yielding $Q_0 \in \mathrm{AUC}_{\mathrm{loc}}$ and rank-monotonicity in the gated, switching regime. |
| KYP-type LMIs for LTV systems with given coefficients (Kalman, 1963; Anderson & Vongpanitlerd, 2013). | A *parametric* LMI in $x$ that must hold uniformly over all admissible selection values, and the derived universal kernel/unobservability and "irreversible forgetting" constraints. |
| Continuous-time analyses of deep SSMs (Halloran et al., 2024; Gu et al., 2022). | A discrete one-step dissipativity inequality and sampled block LMI for the Mamba selective-scan recurrence, used as a differentiable regularizer on a real forecasting architecture (Section 6 and Section 8). |

Table 1: Standard control-theoretic ingredients we leverage, vs. what this paper contributes for selective SSMs.

Concretely, we make the following contributions.

1. **A clean two-signal model.** We formulate continuous-time selective SSMs as port-Hamiltonian-style systems with selection signal $x(\cdot)$ and driving input $u(\cdot)$, under mild $L^p_{\mathrm{loc}}$ regularity that is satisfied by piecewise-constant gating (Section 3).

2. **Forgetting from strict dissipativity.** We show that state-strict dissipativity ($\beta > 0$) plus quadratic bounds on a storage functional implies exponential decay of homogeneous trajectories,

uniformly over admissible selection schedules (Theorem 4.1). This is our formal statement of "forgetting" for selective SSMs.

3. **Frozen-selection structure.** We show that freezing the selection ($x(t) \equiv 0$) yields a genuinely passive LTV input–output subsystem, whose minimal available storage is necessarily quadratic with $AUC_{loc}$ regularity, even under discontinuous gating (Theorem 4.2). This identifies a canonical intrinsic energy structure of selective SSMs and explains why principled initializations such as HiPPO (Gu et al., 2020) matter (Section 4).

4. **Structural consequences of universal quadratic storage.** Under the strong but analytically informative hypothesis that a *single* quadratic storage certifies passivity for *all* admissible selection schedules, we derive a parametric LMI in $x$ (Theorem 5.2) and universal kernel/unobservability conditions on gating, and we formalize "irreversible forgetting": once a state direction is energy-less, structure prevents any selection from making it observable again (Theorem 5.4).

5. **Robust stability under driving inputs.** We give sufficient conditions for global ISS with respect to $u(\cdot)$, uniformly over admissible selection schedules (Theorem 7.1). This is the practically useful counterpart of the structural results.

6. **Continuous-to-discrete bridge for Mamba.** We translate the continuous-time LMI into a one-step discrete dissipativity inequality and a *sampled block LMI* on the Mamba selective-scan recurrence (Section 6). The largest positive eigenvalue of this matrix is differentiable with respect to the model parameters, yielding an explicit Mamba-core regularizer.

7. **Empirical validation on a real Mamba/SST architecture.** On seven standard forecasting datasets (ETTh1/2, ETTm1/2, weather, electricity, traffic) and four prediction horizons, we evaluate the discrete LMI regularizer inside SST (Xu et al., 2025), which uses Mamba as its long-range expert. Across all 28 dataset–horizon pairs we observe roughly 92% reduction in mean Mamba-core LMI violation while clean MSE remains within 0.018% of the unregularized baseline. We also show a conservativeness ablation in which a learnable diagonal storage $P = \text{diag}(q)$ Pareto-improves over $P = I$ in 28/28 pairs at no measurable accuracy cost, and a robustness study under injected perturbations (Gaussian noise, spikes, block masks, amplitude shifts, last-value-hold) where the regularizer improves internal Mamba passivity and state-norm diagnostics but does not by itself produce a robust-MSE improvement (Section 8).

8. **Honest scoping of a router-energy ablation.** We additionally test whether the SST router can learn an *energy-aware* long/short selection schedule by adding a stability-aware penalty. The penalty preserves clean accuracy and consistently nudges the Mamba branch downward, but the train-time correlation between Mamba weight and local LMI violation is essentially zero on aggregate and inconsistent in sign across datasets. We report this as a neutral result and as a concrete pointer to perturbation-aware router training (Section 8, Appendix C.5).

**Reading guide and what is structural vs. practical.** Sections 3–7 are the continuous-time analysis. Section 4 establishes baseline forgetting and the canonical quadratic structure of the frozen-selection subsystem; this is the part that justifies, in energy-based terms, why initializations such as HiPPO are well-behaved. Section 5 examines what happens when a single quadratic storage is required to certify passivity uniformly over selection schedules; the resulting parametric LMI and irreversible-forgetting picture is a strong but useful sufficient lens, not a claim that every Mamba in the wild satisfies it. Section 7 is the practically useful global ISS result. Section 6 connects this to the actual Mamba selective-scan recurrence, and Section 8 reports the empirical evaluation on SST.

We emphasize from the outset that our empirical claim is intentionally narrow. The continuous-time theorems give structural and ISS guarantees under explicit hypotheses. The discrete LMI we derive in Section 6 is a sampled, locally-linearized criterion on the Mamba selective-scan core, not a passivity certificate for the full deep network with normalization, residual connections, and routing. Accordingly, in Section 8 we report what the regularizer actually does: it substantially reduces sampled Mamba-core LMI violations and improves internal stability diagnostics with negligible clean-accuracy cost, and we are explicit about what it does *not*

yet do (e.g., produce a robust-MSE improvement, or induce a strongly energy-aware router schedule). All formal proofs are deferred to Appendix A, the original toy simulations that illustrate the continuous-time theory are in Appendix B, and extended experimental details are in Appendix C.

## 2 Related Work

The analysis of stability and energy-based properties in dynamical systems has a rich history, and most of the technical ingredients we use are classical. Our goal in this section is to make precise which of those ingredients we leverage and where our contribution lies relative to them. The high-level summary is in Table 1 of Section 1.

### 2.1 Dissipativity, Passivity, and LTV Systems

The foundational theory of dissipative systems, pioneered by Willems (1972), provides a general framework for analyzing systems based on energy-like storage functions and supply rates. Passivity, a special case where the supply rate is the input–output inner product, is central to understanding robust stability and interconnection (Van der Schaft, 2000). The Kalman–Yakubovich–Popov (KYP) lemma established a vital link between frequency-domain passivity conditions and state-space properties for LTI systems (Kalman, 1963; Popov, 1961), and was later extended to LTV systems, often involving time-varying Riccati equations or differential/integral inequalities (Anderson & Vongpanitlerd, 2013).

Particularly relevant to our setting is recent work by Morandin & Hinsen (2024), which rigorously investigates quadratic storage functions for passive LTV systems under *minimal regularity assumptions* on the coefficients $(A, B, C, D)$. Two of their results matter for us. First, every passive LTV input–output system in their setting admits a minimal quadratic available storage induced by a matrix function $Q \in \mathrm{AUC}_{\mathrm{loc}}$. Second, this minimal $Q$ has weakly non-increasing rank in time. Our use of these results is specific: we apply them to the *frozen-selection* input–output subsystem of a selective SSM (Section 4), where they immediately yield a canonical intrinsic energy structure of $\mathrm{AUC}_{\mathrm{loc}}$ regularity even when the gating signal $\Delta(t)$ is piecewise-constant. We do not extend this LTV theory, what we add is the recognition that selective SSMs contain such a passive LTV subsystem after we separate the selection signal from the port input, and the structural consequences this has on gating (Section 5).

### 2.2 Stability of Time-Varying and Switched Systems

The input-selectivity of modern SSMs creates dynamics that behave like switched systems, where the "switching signal" is the input data itself. This necessitates tools for analyzing systems with discontinuous parameter changes (Liberzon, 2003), such as methods involving common or multiple Lyapunov functions (Branicky, 1998). The formalisms of Filippov (1988) and Coddington et al. (1956) provide the theoretical bedrock for guaranteeing that solutions to our system exist under the mild $L^p_{\mathrm{loc}}$ regularity we assume, which is, in particular, satisfied by piecewise-constant gating. Our work builds directly on this foundation. We move beyond proving solution existence to ask how a single, coherent energy structure, an $\mathrm{AUC}_{\mathrm{loc}}$ quadratic storage function, can persist across these data-driven switches, and what structural constraints this persistence imposes on the gating mechanism of a selective SSM.

### 2.3 Input-to-State Stability (ISS) for Nonlinear Systems

The ISS framework (Sontag et al., 1989; Sontag & Wang, 1995; Jiang et al., 1999) provides a robust notion of stability for nonlinear systems subject to external inputs, characterized by ISS-Lyapunov functions. It quantifies how the system state is affected by both initial conditions and input magnitudes. While classical ISS theory often assumes smooth dynamics, its principles transfer naturally to selective SSMs after we make the two-signal decomposition explicit: the selection signal $x(\cdot)$ renders the dynamics schedule-dependent, and the port input $u(\cdot)$ enters as the perturbation in the supply rate. Our contribution in this direction (Theorem 7.1) adapts ISS-Lyapunov reasoning by seeking a common quadratic Lyapunov function whose decay condition holds *uniformly* across all admissible selection values, giving a specific and verifiable condition

under which the selective SSM is globally ISS w.r.t. $u(\cdot)$. Conceptually, this is a direct application of standard ISS technology; the work is in stating the right uniform-in-$x$ hypothesis.

### 2.4 Discretization, Sampled-Data, and Mamba-style Selective Scans

A separate strand of control-theoretic work concerns the conditions under which continuous-time stability or passivity properties survive sampling and discretization. For LTV and switched systems, sampled-data passivity has been studied extensively (Liberzon, 2003; Anderson & Vongpanitlerd, 2013). The exact discretization used in Mamba's selective scan (Gu & Dao, 2024) can be viewed as a particular zero-order-hold-style sampling rule applied per token, with input-dependent step size $\Delta(t)$. In Section 6 we use this view to write the Mamba selective-scan recurrence as a one-step affine update, and to derive a sampled block LMI on its parameters that plays the role of the continuous-time LMI from Theorem 5.2. We do not claim novelty in the general principle of sampled-data passivity, what is new is the explicit instantiation for the Mamba selective-scan core and its use as a differentiable training-time regularizer.

### 2.5 Stability and Dynamics of Modern SSMs in Deep Learning

Recent deep learning SSMs such as S4 (Gu et al., 2022), S5 (Smith et al., 2023), and particularly Mamba (Gu & Dao, 2024) (and its variants), have shown remarkable performance, and theoretical analyses are emerging. Halloran et al. (2024) analyzes Mamba's stability via Lyapunov exponents, showing non-positive maximal exponents and inferring robustness to small perturbations. Our work is complementary: instead of an exponent-based diagnosis of an already-trained Mamba, we provide explicit dissipativity-based conditions and a discrete LMI that can be evaluated, and used as a regularizer, during training of a real Mamba-based architecture (Section 6, Section 8). Other works connect selective SSMs to controlled differential equations and rough-path theory to explain expressivity (Kidger et al., 2020; Lyons, 2014; Cirone et al., 2024; Zubic et al., 2025; Zubić et al., 2026). While the present paper takes a different (energy-based, control-theoretic) angle, these works underscore that selective SSMs sit at a rich mathematical intersection, and we view our contribution as another concrete bridge between that mathematics and the design of selective-scan architectures.

## 3 Preliminaries: Selective SSMs and Passivity Framework

This section introduces the continuous-time selective SSMs that are the focus of our analysis, along with the fundamental control-theoretic concepts that underpin our approach. To make the rest of the paper easier to read for an ML-oriented audience, we first introduce the small piece of notation that will recur throughout (Section 3.1), then explain selective SSMs intuitively (Section 3.2) before formalizing them as the system equation 2, and finally give the energy-based definitions (passivity, dissipativity, quadratic storage) that the analysis is built on.

### 3.1 Notation at a Glance

We collect the recurring notation in Table 2. The most important point is the distinction between two signals: $x(\cdot)$ is the *selection (scheduling) signal* that determines the time-varying coefficients $A, B, C$ at each time, while $u(\cdot)$ is the *driving (port) input* that appears in the supply rate $\Re\langle u, y \rangle$. In token-selective architectures, these are typically the same vector ($u \equiv x$), but the analysis is much cleaner when we keep them separate, see the remark after equation 2. For matrices and vectors, $M^H$ denotes the Hermitian (conjugate) transpose; $\|v\|$ is the standard Euclidean norm; $\langle u, v \rangle = v^H u$ is the inner product; and $\mathbf{S}_+^n(\mathbb{C})$ denotes the cone of $n \times n$ Hermitian positive-semidefinite matrices, with $Q \succeq 0$ used as shorthand for $Q \in \mathbf{S}_+^n(\mathbb{C})$.

### 3.2 Selective SSMs, Intuitively

Modern selective architectures such as Mamba (Gu & Dao, 2024), HGRN (Qin et al., 2024), and GLA (Yang et al., 2024) can all be described, at a continuous-time level of abstraction, in two steps:

| Symbol | Meaning |
|---|---|
| $\mathbb{T} \subseteq [0, +\infty)$ | Continuous-time interval on which the system evolves. |
| $h(t) \in \mathbb{C}^N$ | Recurrent state of the selective SSM. |
| $x(t) \in \mathbb{C}^{d_{\text{in}}}$ | *Selection (scheduling) signal* that parametrizes $A, B, C$ via gating. |
| $u(t) \in \mathbb{C}^{d_{\text{in}}}$ | *Driving (port) input*; appears in the supply rate $\Re\langle u, y \rangle$. |
| $y(t) \in \mathbb{C}^{d_{\text{in}}}$ | Output of the selective SSM. |
| $\Delta(t)$ | Auxiliary gating signal modulating $A, B, C$ together with $x(t)$. |
| $A_{\text{eff}}(t), B_{\text{eff}}(t), C_{\text{eff}}(t)$ | Effective time-varying matrices obtained by substituting the current $\Delta(t)$ and $x(t)$ into $A, B, C$ (Section 3.4). |
| $V(t, h)$ | Storage function (energy-like, non-negative). |
| $V_Q(t, h) = \frac{1}{2} h^H Q(t) h$ | Quadratic storage induced by a matrix function $Q : \mathbb{T} \to \mathbf{S}_+^N(\mathbb{C})$. |
| $\beta \geq 0$ | State-strict dissipativity rate; $\beta > 0$ implies intrinsic energy loss. |
| $\text{AUC}_{\text{loc}}$ | Locally absolutely upper semicontinuous matrix functions (Morandin & Hinsen, 2024); supports controlled jumps. |
| $\text{Ker}(Q(t))$ | Kernel of the storage matrix at time $t$ ("energy-less" state directions). |

Table 2: Notation used throughout the paper. The single most important convention is the separation of the selection signal $x(\cdot)$ (which schedules $A, B, C$) from the driving (port) input $u(\cdot)$ (which appears in the supply rate).

   (i) A small neural network reads the current token (or, more generally, the current selection signal $x(t)$) and an auxiliary gating signal $\Delta(t)$, and produces, at every time $t$, a triple of matrices $(A, B, C)\big(\Delta(t), x(t)\big)$. These matrices are not learned per token from scratch but are computed online from the inputs.

   (ii) These data-dependent matrices are then plugged into a continuous-time linear recurrence on a hidden state $h(t)$, so that the recurrence is locally linear in $h$ but globally non-stationary because $(A, B, C)$ change with $x(t)$ and $\Delta(t)$.

This is what makes selective SSMs distinctive: the recurrence is *linear in the state* but *schedule-dependent*, with the schedule chosen by the data. In particular, the gating signal $\Delta(t)$ is typically piecewise-constant (one value per token), so the effective coefficient matrices undergo discontinuous jumps. This is the regime where $\text{AUC}_{\text{loc}}$-style assumptions on the storage matrix become natural, rather than smoothness assumptions.

In practice, the selection signal $x(t)$ and the input that drives the system are often the same vector: a token, treated both as a query for the gating network and as the recurrent input. We will keep the two roles separate at the level of notation, because passivity statements are clearest when "what schedules $A, B, C$" and "what appears in the supply rate" are not conflated. Whenever we want the token-selective regime, we simply specialize to the coupling $u(\cdot) \equiv x(\cdot)$ at the end.

### 3.3 System Definition and Assumptions

We consider dynamical systems evolving in continuous time $t \in \mathbb{T}$, where $\mathbb{T} \subseteq [0, +\infty)$ is a time interval. The internal state of the system at time $t$ is denoted by $h(t) \in \mathbb{C}^N$ (or $\mathbb{R}^N$ in real-valued cases). A key point for our passivity analysis is to distinguish between: (i) a *driving (port) input* $u(t) \in \mathbb{C}^{d_{\text{in}}}$, which is the signal that appears in the supply rate $\Re\langle u, y \rangle$, and (ii) a *selection (scheduling) signal* $x(t) \in \mathbb{C}^{d_{\text{in}}}$ (e.g. the token embedding) that parametrizes the time-varying matrices via input-dependent gating.[1]

The system produces an output $y(t) \in \mathbb{C}^{d_{\text{in}}}$. For matrices and vectors, $M^H$ denotes the Hermitian (conjugate) transpose. The standard Euclidean norm of a vector $v$ is $\|v\|$, and the inner product of two vectors $u, v$ is $\langle u, v \rangle = v^H u$. We denote the space of $n \times n$ Hermitian positive-semidefinite matrices by $\mathbf{S}_+^n(\mathbb{C})$; for $Q \in \mathbf{S}_+^n(\mathbb{C})$, we write $Q \succeq 0$.

---

[1]Throughout Sections 3–7 we assume $d_{\text{out}} = d_{\text{in}}$ so that $\langle u, y \rangle$ and the passivity LMI blocks are dimensionally well-defined; this is the common setting in sequence models where the recurrent channel has matched input/output width.

Our work focuses on Continuous-Time Selective SSMs. Inspired by architectures such as Mamba (Gu & Dao, 2024), their defining characteristic is that the system's core parameters are dynamically modulated by the selection signal $x(t)$ and an auxiliary gating signal $\Delta(t)$. Formally, the dynamics are given by:

$$\begin{cases} \dot{h}(t) = A\Big(\Delta(t), x(t)\Big) h(t) + B\Big(\Delta(t), x(t)\Big) u(t), \\ y(t) = C\Big(\Delta(t), x(t)\Big) h(t). \end{cases} \tag{2}$$

The equation 2 should be read as "a continuous-time abstraction of a selective scan": at each time $t$, the gating mechanism produces $(\Delta(t), x(t))$, these are fed into the per-architecture maps that produce the matrices $A, B, C$, and the resulting linear update acts on the recurrent state $h(t)$ with $u(t)$ as the driving input. We will refer to the substituted matrices $A_{\text{eff}}(t) := A(\Delta(t), x(t))$, $B_{\text{eff}}(t) := B(\Delta(t), x(t))$, and $C_{\text{eff}}(t) := C(\Delta(t), x(t))$ as the *effective* time-varying matrices: they are what the recurrence actually sees once the gating mechanism has produced its outputs at time $t$. The next subsection states the regularity assumption we make on these effective matrices to guarantee well-posedness.

**Remark 3.1** (Coupled-input selective SSMs)**.** In many token-selective architectures, the same signal both *selects* the parameters and *drives* the dynamics; this corresponds to the special case $u(\cdot) \equiv x(\cdot)$. In this paper, we keep $u(\cdot)$ and $x(\cdot)$ conceptually distinct to make the quantifiers in the passivity analysis explicit. Any condition proved for all admissible pairs $(x(\cdot), u(\cdot))$ remains valid (as a sufficient condition) under the coupling $u = x$.

### 3.4 Well-Posedness and Function Space Assumptions

To ensure the system equation 2 is well-defined even with discontinuous parameter changes, we adopt mild regularity conditions. For any admissible input trajectory $x(\cdot)$, let $A_{\text{eff}}(t) := A(\Delta(t), x(t))$ (and similarly for $B_{\text{eff}}, C_{\text{eff}}$). We assume the effective matrices belong to standard Lebesgue spaces: $A_{\text{eff}}(\cdot) \in L^1_{\text{loc}}$, $B_{\text{eff}}(\cdot) \in L^2_{\text{loc}}$, and $C_{\text{eff}}(\cdot) \in L^2_{\text{loc}}$, with the selection signal $x(\cdot) \in L^2_{\text{loc}}$ and the driving input $u(\cdot) \in L^2_{\text{loc}}$. These conditions ensure that the ODE for $h(t)$ satisfies Carathéodory conditions, guaranteeing the local existence and uniqueness of an absolutely continuous solution $h(\cdot) \in W^{1,1}_{\text{loc}}$ (Filippov, 1988; Coddington et al., 1956).

**Remark 3.2** (Feasibility of Regularity Assumptions)**.** These assumptions are mathematically mild and readily satisfied by modern selective SSMs. In token-based architectures such as Mamba, the selection mechanism yields system matrices that are **piecewise-constant**. Such functions are well-behaved members of the $L^p_{\text{loc}}$ spaces, confirming that our framework is grounded in practical implementations while being general enough for future architectures.

### 3.5 Passivity and Dissipativity

We analyze the system equation 2 using the powerful energy-based frameworks of passivity and dissipativity (Willems, 1972). Intuitively, a passive system cannot generate its own energy. It can only store or dissipate energy supplied from its input. A strictly dissipative system is even stronger, as it inherently loses energy over time. These concepts are formalized in the following definitions.

**Definition 3.3** (Storage Function)**.** A function $V : \mathbb{T} \times \mathbb{C}^N \to \mathbb{R}$ is a *storage function* if it is non-negative, i.e., $V(t, h) \geq 0$ for all $(t, h)$, and typically $V(t, 0) = 0$.

**Definition 3.4** (Passivity)**.** The system equation 2 is *passive* if there exists a storage function $V$ such that for all admissible trajectories on any interval $[t_0, T]$:

$$V\big(T, h(T)\big) - V\big(t_0, h(t_0)\big) \leq \int_{t_0}^{T} \text{Re} \langle u(\tau), y(\tau) \rangle \, d\tau. \tag{3}$$

The term $\text{Re} \langle u(\tau), y(\tau) \rangle$ is the instantaneous power, or *supply rate*, provided to the system.

**Definition 3.5** (Strict Dissipativity)**.** The system is *strictly dissipative* with rate $\beta > 0$ if it satisfies the stronger inequality:

$$V\big(T, h(T)\big) - V\big(t_0, h(t_0)\big) \leq \int_{t_0}^{T} \text{Re} \langle u(\tau), y(\tau) \rangle \, d\tau - \beta \int_{t_0}^{T} \|h(\tau)\|^2 \, d\tau. \tag{4}$$

A positive $\beta$ implies an intrinsic rate of energy dissipation, which is crucial for proving strong stability properties like exponential decay. Given the absolute continuity of $h(\cdot)$ and local Lipschitzness of $V$ in $h$, these integral inequalities have corresponding differential forms: $\frac{d}{dt}V \leq \operatorname{Re}\langle u, y\rangle$ for passivity, and $\frac{d}{dt}V \leq \operatorname{Re}\langle u, y\rangle - \beta\|h\|^2$ for strict dissipativity.

### 3.6 Quadratic Storage Functions and AUC Regularity

A significant portion of our analysis focuses on *quadratic storage functions* of the form $V_Q(t, h) = \frac{1}{2}h^H Q(t)h$, where $Q : \mathbb{T} \to \mathbf{S}_+^N(\mathbb{C})$ is a time-varying matrix. The regularity of $Q(t)$ is critical in our setting, as the gating mechanism can induce discontinuities in the system dynamics.

To handle this robustly, we consider $Q(t)$ to belong to the class of Locally Absolutely Upper Semicontinuous matrix functions, denoted $Q \in \operatorname{AUC}_{\text{loc}}(\mathbb{T}, \mathbf{S}_+^N(\mathbb{C}))$, following the framework of (Morandin & Hinsen, 2024). For our purposes, $\operatorname{AUC}_{\text{loc}}$ allows controlled discontinuities and ensures $\dot{Q}(t)$ exists a.e. Under the additional hypothesis that $Q$ induces a universal quadratic storage for a passive LTV subsystem implies further structural constraints, including rank monotonicity (Theorem 5.1). This property is the foundation for the "irreversible forgetting" we analyze later. Formally, a function $Q \in \operatorname{AUC}_{\text{loc}}$ is characterized by being of locally bounded variation ($\operatorname{BV}_{\text{loc}}$), having a derivative $\dot{Q}(t)$ that exists almost everywhere, and satisfying certain integral and jump conditions. The key property that enforces rank monotonicity is that the singular part of its decomposition must be weakly monotonically decreasing in the Loewner order (e.g., at any jump, $\lim_{\tau \to t^-} Q(\tau) \succeq Q(t)$). This mathematical structure is precisely what makes the framework suitable for analyzing switched systems while retaining essential properties about their energy evolution.

## 4 Stability and Structural Properties from Passivity

We establish foundational stability guarantees and explore the core structural properties of passive selective SSMs. We begin by analyzing the *homogeneous frozen-selection* case, where the selection signal is fixed to a reference value $x(t) \equiv 0$ and the port input is turned off $u(t) \equiv 0$. The resulting dynamics $\dot{h}(t) = A(\Delta(t), 0)h(t)$ are determined entirely by the model's initialization, allowing us to isolate and understand its intrinsic stability and memory properties. As we demonstrate in our simulation study in Appendix B.1, this baseline behavior is a powerful predictor of a model's practical performance, and it underpins the empirical observation that the LMI-regularizer introduced in Section 6 preserves clean MSE in our SST/Mamba experiments (Section 8). We first show how strict dissipativity leads to exponential forgetting, and then prove that the unforced dynamics possess a fundamental, well-behaved quadratic energy structure.

### 4.1 Exponential Decay from Strict Dissipativity

The most fundamental stability guarantee arises when the system intrinsically dissipates energy faster than it stores it, even with zero input. This leads to exponential convergence of the state to the origin.

**Theorem 4.1** (Exponential Decay from Strict Dissipativity). Consider the continuous-time selective state-space model defined in Eq. equation 2. Suppose there exists a storage functional $V : [0, \infty) \times \mathbb{C}^N \to \mathbb{R}_{\geq 0}$ satisfying:

(i) **Strict Dissipativity Inequality:** For some constant $\beta > 0$, every admissible pair $(x(\cdot), u(\cdot))$ and the corresponding state–input–output trajectory $\{h(\tau), u(\tau), y(\tau)\}$ on any interval $[t_0, T]$ satisfies:

$$V\big(T, h(T)\big) - V\big(t_0, h(t_0)\big) \leq \int_{t_0}^{T} \operatorname{Re}\langle u(\tau), y(\tau)\rangle \, d\tau - \beta \int_{t_0}^{T} \|h(\tau)\|^2 \, d\tau. \tag{5}$$

(ii) **Regularity and Quadratic Bounds:** $V(t, h)$ is locally Lipschitz in $h$, absolutely continuous in $t$ along system trajectories, and there exist constants $k_2 \geq k_1 > 0$ such that for all $t \geq 0$ and $h \in \mathbb{C}^N$:

$$k_1\|h\|^2 \leq V(t, h) \leq k_2\|h\|^2. \tag{6}$$

Then, for any admissible selection schedule $x(\cdot)$, the corresponding homogeneous (zero port-input) trajectory (i.e., when $u(t) \equiv 0$ for $t \geq 0$) exhibits exponential decay: there exist constants $C \geq 1$ and $\gamma > 0$ such that for any initial state $h(0)$, the solution satisfies:

$$\|h(t)\| \leq C\, e^{-\gamma t}\, \|h(0)\| \quad \text{for all } t \geq 0. \tag{7}$$

*Proof Sketch.* The proof considers the system without port input, i.e., $u = 0$ (so the supply term $\text{Re}\,\langle u, y \rangle$ vanishes). In this case, the strict dissipativity inequality simplifies to $\frac{dV}{dt} \leq -\beta\|h\|^2$. Since the energy function $V$ is assumed to be quadratically bounded, satisfying $k_1\|h\|^2 \leq V \leq k_2\|h\|^2$, this implies $\frac{dV}{dt} \leq -\gamma V$ for some constant $\gamma > 0$. This is a classic differential inequality whose solution is an exponential decay. The bounds on $V$ then directly translate this exponential decay of energy into an exponential decay of the state norm $\|h(t)\|$. Check Appendix A.1 for the full proof. $\qquad \square$

**ML implication.** *For a selective SSM, this is the formal statement of "forgetting": as long as the architecture admits any storage functional $V$ with strict dissipativity ($\beta > 0$) and quadratic bounds, the recurrent state decays exponentially in the absence of external drive, irrespective of which selection schedule $x(\cdot)$ the gating produces. Whether such a $V$ exists is a property of the architecture and its initialization; the storage functional we will exhibit in Section 4.2 (and use as the reference energy for HiPPO-style initialization in Appendix B.1) is one concrete witness.*

## 4.2 Inherent Quadratic Structure and Regularity in Frozen-Selection LTV Dynamics

While Theorem 4.1 guarantees stability under strict dissipativity, it does not specify the form of the storage function $V$. We now connect selective SSMs to the passive LTV framework of Morandin & Hinsen (2024). The key point is that the results of Morandin & Hinsen (2024) apply to *input–output* LTV systems of the form

$$\dot{h}(t) = A(t)h(t) + B(t)u(t), \qquad y(t) = C(t)h(t) + D(t)u(t), \tag{8}$$

with mild regularity on $(A, B, C, D)$ and passivity defined via the supply rate $\Re\,\langle u, y \rangle$ (Morandin & Hinsen, 2024, Eq. (1), Def. 1.1). Accordingly, instead of removing the input channel, we *freeze the selection signal* (e.g., set $x(t) \equiv 0$) and obtain a genuine LTV input–output subsystem with port input $u(\cdot)$. For passive LTV systems, the available storage is finite and coincides with a minimal quadratic storage $V_a(t, h) = \frac{1}{2}h^H Q(t)h$ with $Q \in \text{AUC}_{\text{loc}}$ (Morandin & Hinsen, 2024, Def. 4.1, Cor. 4.3). Evaluating this quadratic storage along homogeneous (zero port-input) trajectories $u \equiv 0$ yields a canonical quadratic energy functional for the frozen-selection baseline dynamics (i.e., $x \equiv 0$ and $u \equiv 0$).

To prove this, we leverage the concept of the *available storage* function, which represents the maximum energy that can be extracted from the system from a given state. It is a standard result in dissipativity theory that if any storage function exists, then the available storage function is well-defined and is itself a valid storage function (see Appendix A.2). The following theorem connects this concept to the structure of selective SSMs.

**Theorem 4.2** (Existence and $\text{AUC}_{\text{loc}}$ regularity of a minimal quadratic storage for the frozen-selection LTV subsystem). Consider the selective SSM equation 2 with selection signal $x(\cdot)$ and driving input $u(\cdot)$. Assume the coefficient maps satisfy the standard well-posedness regularity: for admissible $x(\cdot)$ we have

$$A(\Delta(\cdot), x(\cdot)) \in L^1_{\text{loc}}(\mathbb{T}), \quad B(\Delta(\cdot), x(\cdot)) \in L^2_{\text{loc}}(\mathbb{T}), \quad C(\Delta(\cdot), x(\cdot)) \in L^2_{\text{loc}}(\mathbb{T}),$$

and $u(\cdot) \in L^2_{\text{loc}}(\mathbb{T})$. (Here $D \equiv 0$, hence $D \in L^\infty_{\text{loc}}$ holds trivially.)

Suppose there exists a storage functional $V : \mathbb{T} \times \mathbb{C}^N \to \mathbb{R}_{\geq 0}$ with $V(t, 0) = 0$ such that for some $\beta \geq 0$ the dissipativity inequality

$$V\big(T, h(T)\big) - V\big(t_0, h(t_0)\big) \leq \int_{t_0}^{T} \text{Re}\,\langle u(\tau), y(\tau) \rangle\, d\tau - \beta \int_{t_0}^{T} \|h(\tau)\|^2\, d\tau \tag{9}$$

holds for all admissible pairs $(x(\cdot), u(\cdot))$ and all corresponding trajectories of equation 2. (In particular, dropping the nonpositive term $-\beta \int \|h\|^2$ yields the standard passivity inequality with supply rate $\Re\,\langle u, y \rangle$.)

Fix the *frozen-selection* signal $x(t) \equiv 0$ and define the associated LTV input–output system

$$\begin{cases} \dot{h}(t) = A_0(t)h(t) + B_0(t)u(t), \\ y_0(t) = C_0(t)h(t), \end{cases} \quad \text{where} \quad A_0(t) = A(\Delta(t), 0), \ B_0(t) = B(\Delta(t), 0), \ C_0(t) = C(\Delta(t), 0).$$
(10)

Then the following hold:

(a) **Passivity of the frozen-selection LTV system.** The LTV system equation 10 is passive in the sense of Morandin & Hinsen (2024, Def. 1.1). Indeed, restricting equation 9 to $x(\cdot) \equiv 0$ yields a dissipation inequality for every state–input–output solution $(h, u, y_0)$ of equation 10.

(b) **Existence of minimal quadratic available storage.** Let $V_{a,0}$ denote the available storage of equation 10 in the sense of Morandin & Hinsen (2024, Def. 4.1). Since equation 10 is passive and satisfies the regularity assumptions of Morandin & Hinsen (2024, Eq. (1)), its available storage is finite and coincides with a quadratic storage,

$$V_{a,0}(t, h) = \tfrac{1}{2} h^H Q_0(t) h,$$

for some matrix function $Q_0 : \mathbb{T} \to \mathbf{S}_+^N(\mathbb{C})$, and $V_{a,0}$ is minimal among all storage functions (Morandin & Hinsen, 2024, Cor. 4.3).

(c) $\mathrm{AUC}_{\mathrm{loc}}$ **regularity of $Q_0(t)$.** The inducing matrix function can be chosen with $\mathrm{AUC}_{\mathrm{loc}}$ regularity:

$$Q_0 \in \mathrm{AUC}_{\mathrm{loc}}(\mathbb{T}, \mathbf{S}_+^N(\mathbb{C})),$$

again by Morandin & Hinsen (2024, Cor. 4.3).

Finally, the homogeneous (unforced) dynamics of the frozen-selection system correspond to setting $u \equiv 0$ in equation 10, i.e., $\dot{h} = A_0(t)h$. If, in addition, $V$ satisfies the hypotheses of Theorem 4.1 (strict dissipativity $\beta > 0$ and quadratic bounds), then these homogeneous trajectories decay exponentially as stated in Theorem 4.1.

*Proof Sketch.* The frozen-selection system equation 10 is an LTV input–output system of the form

$$\dot{h}(t) = A_0(t)h(t) + B_0(t)u(t), \qquad y_0(t) = C_0(t)h(t) + D_0(t)u(t),$$

with $D_0 \equiv 0$. Under our assumptions we have $A_0 \in L_{\mathrm{loc}}^1$, $B_0, C_0 \in L_{\mathrm{loc}}^2$, and $D_0 \in L_{\mathrm{loc}}^\infty$, hence equation 10 fits the setting of Morandin & Hinsen (2024, Eq. (1)).

Since equation 9 holds for all admissible $(x(\cdot), u(\cdot))$, it holds in particular when restricting to $x(\cdot) \equiv 0$. Dropping the nonpositive term $-\beta \int \|h\|^2$ yields the passivity dissipation inequality of Morandin & Hinsen (2024, Def. 1.1) for equation 10, so the frozen-selection LTV system is passive.

For passive LTV systems, Morandin & Hinsen (2024, Cor. 4.3) shows that the available storage (Morandin & Hinsen, 2024, Def. 4.1) is finite and coincides with a minimal quadratic storage $V_{a,0}(t, h) = \tfrac{1}{2} h^H Q_0(t) h$ with $Q_0 \in \mathrm{AUC}_{\mathrm{loc}}$. Finally, the homogeneous dynamics correspond to fixing $u \equiv 0$ (cf. Morandin & Hinsen (2024, Sec. 2.1)), and exponential decay under strict dissipativity follows from Theorem 4.1. □

**ML implication.** *This is the canonical intrinsic energy of a selective SSM at its initialization. Once the gating mechanism produces piecewise-constant matrix updates ($\Delta(t)$ is piecewise-constant per token), the matrix $Q_0(t)$ that measures "how much state has been usefully stored" is in $\mathrm{AUC}_{\mathrm{loc}}$: it can jump, but only in controlled (Loewner-monotone-decreasing) ways. This is exactly the mathematical regularity needed for principled initializations such as HiPPO (Gu et al., 2020) to behave well as a baseline energy landscape on which the input-driven selection then operates. Empirically, we observe in Appendix B.1 that initializations admitting such a well-conditioned $Q_0$ (HiPPO-style) yield slow, stable energy decay under input removal, while initializations failing this admit no valid $Q_0$ and diverge. The discrete-time analogue of this storage matrix becomes the matrix $P$ that we regularize in Section 6 and Section 8.*

# 5 Constraints Imposed by Universal Quadratic Passivity on Gating Mechanisms

Building on the foundational properties of the unforced system, this section studies the inherent design constraints of selective SSMs. We investigate the consequences of a strong but informative assumption: that a *single*, universal quadratic storage function $V_Q(t, h)$ guarantees passivity across all possible selection-driven (schedule-dependent) dynamics. The power of this assumption lies not in its generality, but in the strict, necessary consequences it imposes on the gating mechanism. This approach allows us to derive concrete constraints in the form of LMIs and to formalize a notion of "irreversible forgetting".

We emphasize from the outset that the universal-storage hypothesis is a *sufficient* structural condition: it is what we use to expose what passivity "would have to" force on a selective SSM if a single quadratic energy were sufficient to certify it for every gating decision. We do *not* claim that every Mamba in the wild satisfies it; rather, the hypothesis isolates a clean structural picture (a parametric LMI in $x$, kernel/unobservability constraints, and irreversible forgetting) which then motivates, in Section 6, the discrete LMI we use as a differentiable training-time regularizer on a real Mamba-based architecture, where the LMI is enforced as a soft constraint rather than as a fact about the architecture.

## 5.1 Regularity and Rank Monotonicity of Universal Quadratic Storage

If a single quadratic function $V_Q$ is capable of certifying passivity regardless of the input-driven variations in system parameters, the matrix $Q(t)$ defining this function must possess inherent structural regularity and obey specific monotonicity properties.

**Theorem 5.1** (Regularity and Rank of Universal Quadratic Storage)**.** Consider the continuous-time selective SSM equation 2 under the standard regularity assumptions. Suppose there exists a time-varying quadratic storage function $V_Q(t, h) = \frac{1}{2} h^H Q(t) h$ (with $Q : \mathbb{T} \to \mathbf{S}_+^N(\mathbb{C})$) that satisfies the dissipativity inequality (which implies passivity by dropping the nonpositive term):

$$V_Q\big(T, h(T)\big) - V_Q\big(t_0, h(t_0)\big) \le \int_{t_0}^{T} \mathrm{Re}\, \langle u(\tau), y(\tau) \rangle\, d\tau - \beta \int_{t_0}^{T} \|h(\tau)\|^2\, d\tau \tag{11}$$

for some $\beta \ge 0$, valid for *all* admissible pairs $(x(\cdot), u(\cdot))$ and all corresponding state–input–output trajectories $\{h(\tau), u(\tau), y(\tau)\}$ of equation 2. Then, the matrix function $Q(t)$ must satisfy the following properties:

(a) **AUC Regularity:** $Q(t)$ must belong to the class of locally absolutely upper semicontinuous matrix functions, i.e., $Q \in \mathrm{AUC}_{\mathrm{loc}}(\mathbb{T}, \mathbf{S}_+^N(\mathbb{C}))$.

(b) **Rank Monotonicity:** The rank of $Q(t)$, denoted $r(t) = \mathrm{rank}(Q(t))$, must be weakly monotonically non-increasing over time $t \in \mathbb{T}$.

*Proof Sketch.* The logic here is similar to the previous theorem but applied to the storage matrix $Q(t)$ itself rather than the minimal one. If a single quadratic function $V_Q$ works for all admissible pairs $(x(\cdot), u(\cdot))$, then in particular it works for the frozen-selection LTV subsystem obtained by fixing $x(\cdot) \equiv 0$. Hence $V_Q$ is a valid quadratic storage function for that passive LTV system, and the structural results of Morandin & Hinsen (2024) imply $Q \in \mathrm{AUC}_{\mathrm{loc}}$ and that $t \mapsto \mathrm{rank}(Q(t))$ is weakly non-increasing. The same theoretical results from Morandin & Hinsen (2024) that impose structure on the minimal storage function also apply to any valid quadratic storage function. This forces $Q(t)$ to have $\mathrm{AUC}_{\mathrm{loc}}$ regularity and, crucially, a non-increasing rank over time. Check Appendix A.4 for the full proof. □

**ML implication.** *For a selective SSM, this says: if there is any single quadratic "energy meter" $V_Q$ that the architecture cannot beat regardless of the data-dependent gating decisions, then the meter itself must obey two structural laws. It is allowed to jump (gating produces piecewise-constant updates), but only in $\mathrm{AUC}_{\mathrm{loc}}$ ways, and the rank of the matrix $Q(t)$ that defines it cannot grow over time. This is the precise sense in which a selective SSM, viewed through a single passivity certificate, can only ever lose dimensions of usable energy storage. The next two theorems unpack what this rank-non-increasingness means operationally for gating.*

## 5.2 Parametric LMI and Kernel Constraints on Gating

The existence of a universal quadratic storage function $V_Q$ translates into concrete algebraic constraints that the selection-dependent matrices $A(t,x), B(t,x), C(t,x)$ must satisfy *parametrically* for all admissible *selection values* $x$. Equivalently, for each fixed selection value $x$ and almost every time $t$, the resulting quadratic dissipation inequality must hold for all states $h$ and all port inputs $u$. These constraints take the form of a parametric Linear Matrix Inequality (LMI) and lead to important conditions on how the output matrix $C(t,x)$ interacts with the kernel of the storage matrix $Q(t)$.

**Theorem 5.2** (Passivity Constraints on Selection via a Parametric LMI)**.** Consider the selective SSM equation 2 with selection signal $x(\cdot)$ and driving input $u(\cdot)$ under the standard regularity assumptions. Assume the system's passivity is guaranteed by a *single*, time-varying quadratic storage function

$$V_Q(t,h) = \frac{1}{2}h^H Q(t)h, \qquad Q : \mathbb{T} \to \mathbf{S}_+^N(\mathbb{C}),$$

such that for some $\beta \geq 0$ the dissipativity inequality

$$V_Q\big(T, h(T)\big) - V_Q\big(t_0, h(t_0)\big) \leq \int_{t_0}^{T} \mathrm{Re}\,\langle u(\tau), y(\tau)\rangle \, d\tau - \beta \int_{t_0}^{T} \|h(\tau)\|^2 \, d\tau \tag{12}$$

holds for *all* initial states and for *all* admissible pairs $(x(\cdot), u(\cdot))$.

Then, for almost every $t \in \mathbb{T}$ and for every admissible selection value $x \in \mathbb{C}^{d_{\mathrm{in}}}$, the following parametric LMI holds:

$$\mathcal{L}(t,x) := \begin{bmatrix} \dot{Q}(t) + Q(t)A(t,x) + A(t,x)^H Q(t) + 2\beta I & Q(t)B(t,x) - C(t,x)^H \\ B(t,x)^H Q(t) - C(t,x) & 0 \end{bmatrix} \preceq 0, \tag{13}$$

where $A(t,x) = A(\Delta(t), x)$, $B(t,x) = B(\Delta(t), x)$, $C(t,x) = C(\Delta(t), x)$, and $\dot{Q}(t)$ exists a.e.

In particular, the following universal kernel condition must hold:

$$\forall v \in \mathrm{Ker}\big(Q(t)\big), \quad C\big(\Delta(t), x\big)v = 0 \quad \text{for all admissible } x \text{ at time } t, \text{ a.e. } t \in \mathbb{T}. \tag{14}$$

**Remark 5.3** (Coupled case $u = x$)**.** If the architecture uses a single signal (i.e., $u(\cdot) \equiv x(\cdot)$), then equation 13 remains a strong *sufficient* condition for passivity and motivates the regularizer used in Appendix B.3. Necessity holds under the stronger schedule-independent hypothesis stated in the theorem (passivity for all admissible pairs $(x(\cdot), u(\cdot))$).

*Proof Sketch.* This proof translates the differential form of the passivity inequality, $\frac{dV_Q}{dt} \leq \ldots$, into a matrix inequality. By substituting the system dynamics and the quadratic form of $V_Q$, we arrive at an expression that must be nonpositive for all states $h$ and port inputs $u$, for each fixed selection value $x$ (and for a.e. time $t$). Such an expression can be elegantly represented as a Linear Matrix Inequality (LMI). For the LMI to hold for all $h$ and $u$ (uniformly over admissible selection values $x$), specific conditions must be met. By choosing a state $v$ from the kernel of $Q$ (where $Qv = 0$), the LMI simplifies dramatically, revealing that the term involving the output matrix $C$ must be zero to prevent the inequality from being violated. This leads to the necessary condition that $C(t,x)v = 0$. Check Appendix A.5 for the full proof. $\qquad\square$

**ML implication.** *Equation equation 13 is the formal statement of the practical regularizer we use later. It is a parametric LMI: it must hold not just at one selection value, but uniformly over every selection value $x$ that the gating network can produce at time $t$. This is exactly what we sample and penalize during training of a real Mamba in Section 6: each token produces a selection-dependent matrix $\mathcal{L}(t,x)$, and we drive its largest positive eigenvalue toward zero. The kernel condition equation 14 adds a striking structural reading: any state direction the storage matrix considers "energy-less" (in $\mathrm{Ker}\,Q(t)$) must be unobservable through $C(t,x)$, regardless of which selection value $x$ the gate emits. In Mamba terms, the gating cannot decide to "read out" a direction that the energy structure has already discarded.*

### 5.3 Irreversible Forgetting and Energy Consistency in Kernel Subspace

The constraints derived above, particularly the non-increasing rank of $Q(t)$ and the universal kernel condition for $C(t, x)$, suggest a form of structural irreversibility associated with the modes that fall into the kernel of the storage matrix $Q(t)$. We now explore this concept of "irreversible forgetting" from the perspective of the universal storage function $V_Q$ and examine the energy balance required within this forgotten subspace.

**Theorem 5.4** (Inertness of Forgotten Modes and Gating Constraints from Passivity)**.** Consider the continuous-time selective state-space model equation 2 under standard regularity assumptions. Assume its passivity is guaranteed by a *single*, time-varying quadratic storage function $V_Q(t, h) = \frac{1}{2} h^H Q(t) h$ with $Q : \mathbb{T} \to \mathbf{S}_+^N(\mathbb{C})$, satisfying the dissipativity inequality equation 12 for some $\beta \geq 0$ and for *all* admissible pairs $(x(\cdot), u(\cdot))$. Then, the following properties regarding "forgotten modes" (states in $\mathrm{Ker}(Q(t))$) hold:

1. **Irreversibility in storage dimension (rank monotonicity):** By Theorem 5.1, the matrix function $Q$ can be chosen in $\mathrm{AUC}_{\mathrm{loc}}(\mathbb{T}, \mathbf{S}_+^N(\mathbb{C}))$ and the rank $r(t) = \mathrm{rank}(Q(t))$ is weakly monotonically non-increasing over time. Equivalently, since $Q(t) \in \mathbb{C}^{N \times N}$, $\dim \mathrm{Ker}(Q(t)) = N - r(t)$ is weakly non-decreasing.

2. **Energy Consistency within the Kernel:** For almost every $t \in \mathbb{T}$ where $\dot{Q}(t)$ exists, every direction $v \in \mathrm{Ker}(Q(t))$ satisfies the kernel energy inequality

$$v^H \dot{Q}(t) v + 2\beta \|v\|^2 \leq 0. \tag{15}$$

This ensures the evolution $\dot{Q}(t)$ within the kernel is consistent with the required dissipation $\beta$. If $\beta = 0$, $h^H \dot{Q} h \leq 0$; if $\beta > 0$, $h^H \dot{Q} h \leq -2\beta \|h\|^2 < 0$ (for $h \neq 0$).

3. **Constraint on selection schedules:** Under the universal passivity hypothesis, the parametric LMI equation 13 must hold for almost every $t$ and for every admissible selection value $x$. Hence any selection schedule $x(\cdot)$ that attains values for which equation 13 fails on a set of times of positive measure would contradict the hypothesis. The dynamics must respect the energy structure defined by $Q(t)$ and $\dot{Q}(t)$ within the kernel.

**Remark 5.5** (On strict dissipativity and rank-deficient storage)**.** If $\beta > 0$ and $Q(t) \succeq 0$, then the kernel inequality $v^H \dot{Q}(t) v + 2\beta \|v\|^2 \leq 0$ implies that any nonzero $v \in \mathrm{Ker}(Q(t))$ would satisfy $v^H \dot{Q}(t) v < 0$ at times where $\dot{Q}(t)$ exists, which would contradict the requirement $Q(t + \varepsilon) \succeq 0$ for small $\varepsilon > 0$. Thus, under state-strict dissipativity ($\beta > 0$), $Q(t)$ must be positive definite for almost every $t$, so the kernel-based "irreversible forgetting" discussion is primarily meaningful in the passive case $\beta = 0$.

*Proof Sketch.* This theorem's proof is a direct consequence of the preceding results.

1. **Kernel Non-Shrinking:** This follows immediately from the rank monotonicity established in Theorem 5.1; if the rank cannot increase, the dimension of the kernel ($N - \mathrm{rank}$) cannot decrease.

2. **Energy Consistency:** This is derived by taking the (1,1) block of the LMI from Theorem 5.2 and evaluating it for a state $h$ in the kernel of $Q$. This isolates the term $h^H \frac{dQ}{dt} h$ and shows it must be negative enough to account for the required energy dissipation.

3. **Constraint on Dynamics:** This is a logical conclusion: any selection signal $x$ that would induce dynamics violating the LMI contradicts the fundamental assumption of universal passivity, and is therefore inadmissible.

Check Appendix A.6 for the full proof. $\qquad\square$

**ML implication.** *Read in the passive case ($\beta = 0$, per the Remark above), this is the "irreversible forgetting" picture: once a state direction enters $\mathrm{Ker}\,Q(t)$, the kernel cannot shrink, the LMI prevents any selection from making that direction observable through $C(t, x)$, and the energy structure within the kernel must continue to dissipate consistently. For a Mamba-style selective SSM, this offers a structural lens on what gating "cannot undo" under a single passive certificate, and it foreshadows, on the practical side, why a discrete LMI regularizer monotonically reduces violations across training without the model recovering them*

*later (Section 8, Appendix C). We do not claim every selective SSM in the wild lives in this strong universal-storage regime; rather, the lens identifies the structural bias that the discrete LMI of Section 6 encourages a real Mamba to acquire during training.*

# 6 From Continuous Theory to Discrete Mamba

The continuous-time results of Sections 4–7 give us a clear structural picture: under appropriate dissipativity assumptions, selective SSMs must satisfy a parametric LMI in the selection value $x$, with kernel/unobservability and rank-monotonicity consequences. In practice, however, modern selective architectures such as Mamba (Gu & Dao, 2024) do not run a continuous-time ODE; they run a discrete-time selective scan with an input-dependent step size $\Delta_k$ per token. To make the theory actionable for these architectures, this section translates the LMI of Theorem 5.2 into a sampled-data inequality on the Mamba selective-scan core, and into a differentiable training-time regularizer that we will evaluate empirically in Section 8.

We do not claim novelty in the general principle of sampled-data passivity (Liberzon, 2003; Anderson & Vongpanitlerd, 2013). What is new here is the explicit instantiation of the Mamba selective-scan recurrence and its integration into the training loop of a real architecture. Throughout, we work in the coupled-input regime $u_k \equiv x_k$ (token, gate, and drive collapse into one signal), since this is the regime real Mamba uses; under that coupling, Theorem 5.2 gives a strong sufficient condition for one-step dissipativity rather than a necessity statement, and we use it accordingly.

## 6.1 Discrete Selective-Scan Dynamics

In the standard Mamba selective scan, at each token index $k \in \{0, 1, \dots, T-1\}$ the model produces from the current input $u_k \in \mathbb{C}^{d_{\text{in}}}$ a token-dependent step size $\Delta_k \in \mathbb{R}_{>0}$ and continuous-time matrices $A(\Delta_k, u_k)$, $B(\Delta_k, u_k)$, $C(\Delta_k, u_k)$, then discretizes the per-token continuous update via a zero-order-hold-style rule. Concretely, this yields a discrete recurrence

$$\begin{cases} h_{k+1} = F_k \, h_k + G_k \, u_k, \\ y_k = C_k \, h_{k+1} + D_k \, u_k, \end{cases} \qquad k = 0, 1, \dots, T-1, \tag{16}$$

where the per-token matrices $(F_k, G_k, C_k, D_k)$ are computed from $(\Delta_k, u_k)$ by the architecture's discretization rule. We stress two structural facts about equation 16:

(i) For each fixed $k$, equation 16 is an affine update in $(h_k, u_k)$ with parameters $(F_k, G_k, C_k, D_k)$ that can be read off the Mamba selective-scan code path. In our experiments, these are exactly the per-step quantities that the Mamba block computes during the forward pass.

(ii) The token-to-token dependence of $(F_k, G_k, C_k, D_k)$ on $u_k$ is the discrete counterpart of the continuous-time selectivity of $(A, B, C)$ on $x(t)$. The selection structure of the recurrence is preserved under sampling.

We take equation 16 as our discrete model of the Mamba selective-scan core and develop a passivity criterion on its parameters.

## 6.2 One-Step Dissipativity Inequality

Following the continuous-time framework, we ask whether equation 16 admits a quadratic storage that decays under the supply rate, sampled at the token grid. Let $P \in \mathbf{S}_+^N(\mathbb{C})$ be a positive-semidefinite matrix and define the sampled storage

$$V(h) = \tfrac{1}{2} h^\top P h.$$

A natural sampled analog of strict dissipativity is the requirement that, for every $k$ and every $(h_k, u_k)$,

$$V(h_{k+1}) - V(h_k) \leq u_k^\top y_k - \beta \|h_k\|^2, \tag{17}$$

for some $\beta \geq 0$. Inequality equation 17 is the discrete analog of the differential dissipativity bound used in the proof of Theorem 5.2.

**Proposition 6.1** (Sampled block LMI for the Mamba selective-scan core)**.** For the discrete recurrence equation 16 with quadratic storage $V(h) = \frac{1}{2}h^\top P h$ and dissipation parameter $\beta \geq 0$, the one-step dissipativity inequality equation 17 holds for every $(h_k, u_k)$ if and only if the per-step block matrix

$$\mathcal{L}_k(P, \beta) \; \coloneqq \; \begin{bmatrix} F_k^\top P F_k - P + 2\beta I & F_k^\top P G_k - F_k^\top C_k^\top \\ G_k^\top P F_k - C_k F_k & G_k^\top P G_k - C_k G_k - G_k^\top C_k^\top - 2D_k \end{bmatrix} \; \preceq \; 0 \tag{18}$$

is negative semidefinite for that token. The matrix $\mathcal{L}_k(P, \beta)$ is differentiable in the model parameters that produce $(F_k, G_k, C_k, D_k)$ and in $P$.

*Proof Sketch.* Substitute $h_{k+1} = F_k h_k + G_k u_k$ into $V(h_{k+1})$ and $y_k = C_k h_{k+1} + D_k u_k = C_k F_k h_k + (C_k G_k + D_k) u_k$ into the supply rate $u_k^\top y_k$. The inequality equation 17 becomes a quadratic form in the augmented vector $z_k = [h_k^\top, \, u_k^\top]^\top$:

$$z_k^\top \mathcal{L}_k(P, \beta) z_k \; \leq \; 0 \quad \text{for all } z_k,$$

which is equivalent to $\mathcal{L}_k(P, \beta) \preceq 0$. Differentiability in $(F_k, G_k, C_k, D_k, P)$ is immediate from the polynomial structure of the entries. $\qquad\qquad\qquad\qquad\qquad\qquad\qquad\qquad\qquad\qquad\qquad\qquad\qquad\qquad\qquad\qquad\qquad\quad\square$

**Reading.** Proposition 6.1 is exactly the discrete sampled instance of the parametric LMI in Theorem 5.2: the role of $Q(t)$ is played by the constant matrix $P$, the role of $\dot{Q}(t) + QA + A^\top Q$ is played by $F_k^\top P F_k - P$, and the off-diagonal $QB - C^\top$ is replaced by $F_k^\top P G_k - F_k^\top C_k^\top$. Forcing $\mathcal{L}_k(P, \beta) \preceq 0$ at every token is the discrete analog of forcing the parametric LMI to hold pointwise in $x$.

**Scope of the criterion.** A few cautious words about what equation 18 does and does not say. First, $(F_k, G_k, C_k, D_k)$ are local linearizations of the Mamba selective-scan core at token $k$; they are not the parameters of the full deep network with normalization, residual connections, and routing. Imposing $\mathcal{L}_k(P, \beta) \preceq 0$ is therefore a structural statement about the Mamba core, not a passivity certificate for the surrounding architecture. Second, in line with our coupled-input framing, equation 18 is a strong sufficient condition for the one-step dissipativity inequality equation 17: once the LMI holds at every $k$, the storage $V(h)$ is non-increasing along trajectories driven by $u_k \equiv x_k$ up to the supply term, regardless of the gating rule that produced $(\Delta_k, u_k)$. Third, the matrix $P$ is a design choice; the simplest case $P = I$ already gives a useful (Euclidean) energy meter, and we will see in Section 8 that learnable diagonal $P = \text{diag}(q)$ produces a small but consistent Pareto improvement.

### 6.3 Practical Mamba-Core regularizer

Proposition 6.1 suggests a clean training-time mechanism. Given a real Mamba forward pass on a batch of tokens, we can read off $(F_k, G_k, C_k, D_k)$ at every step (they are exactly the per-token matrices used in the selective scan), choose a storage matrix $P$ and a dissipation parameter $\beta \geq 0$, form the per-step block matrix $\mathcal{L}_k(P, \beta)$, and penalize its largest positive eigenvalue. Concretely, define

$$\mathcal{R}_{\text{LMI}}(\theta; P, \beta) \; \coloneqq \; \mathbb{E}_k \big[ \max\big(0, \, \lambda_{\max}(\mathcal{L}_k(P, \beta))\big) \big], \tag{19}$$

where $\theta$ collects the trainable parameters that determine $(F_k, G_k, C_k, D_k)$ in the Mamba core, and the expectation is taken over tokens (and optionally Mamba channels). The total training loss is then

$$\mathcal{L}_{\text{total}}(\theta) \; = \; \mathcal{L}_{\text{task}}(\theta) \; + \; \lambda_{\text{LMI}} \, \mathcal{R}_{\text{LMI}}(\theta; P, \beta),$$

with $\lambda_{\text{LMI}} \geq 0$ a single weight. We will refer to $\mathcal{R}_{\text{LMI}}$ as the *discrete LMI regularizer*.

A few practical remarks. Computing $\lambda_{\max}(\mathcal{L}_k(P, \beta))$ on small per-token blocks is cheap: in the architectures we consider in Section 8, $\mathcal{L}_k(P, \beta)$ is at most a few tens of dimensions, and its largest eigenvalue (with autograd-friendly subgradients) is computed via standard symmetric eigendecomposition. The choice of $P$ controls how conservative the criterion is. We use $P = I$ as our main practical method and report a learnable

diagonal $P = \text{diag}(q)$ as an ablation; we found that the optional projection step that re-PSDifies $P$ after each update is essentially inactive in our runs, so we do not present it as a central component. Finally, the regularizer is purely token-local at the level of the Mamba core; it does not require backpropagating through the entire sequence to evaluate, and its computational cost is negligible compared with the Mamba forward pass.

**What this section does and does not assert.** The reader should take three things from this bridge. (i) Proposition 6.1 is the discrete, sampled-data form of the continuous parametric LMI of Theorem 5.2, instantiated explicitly for the Mamba selective-scan recurrence. (ii) The criterion is a Mamba-core property: it constrains the per-step matrices that the Mamba block produces, not the surrounding deep network. (iii) The criterion is differentiable, so it can be used as a soft constraint during training. Whether enforcing it in this way actually changes how a real Mamba behaves — on clean test data, on input perturbations, and on routing-energy signals — is an empirical question, which we now answer in Section 8 on seven standard time-series datasets, with extended detail in Appendix C.

## 7 Robust Stability under Input-Driven Dynamics

Sections 4–5 laid out the structural side of selective SSMs — forgetting from strict dissipativity, the canonical quadratic structure of the frozen-selection subsystem, and the parametric-LMI consequences of universal quadratic storage. Section 6 then translated those structural results into a sampled, differentiable criterion for the Mamba selective-scan core. We now return to the continuous-time setting and analyze the selective SSM in the fully general and realistic case where it is driven by persistent, non-zero *port inputs* $u(\cdot)$ under an exogenous selection schedule $x(\cdot)$. This section provides sufficient conditions for Input-to-State Stability (ISS), the gold standard for robust stability in nonlinear systems. To achieve this strong guarantee, we must impose conditions that ensure the system's internal dynamics are powerful enough to overcome any disturbance from the input. These conditions, while strong, provide a clear, verifiable pathway to designing certifiably robust models and are the continuous-time counterparts of the practical regularizer used in Section 8.

### 7.1 Global ISS from Uniform Dissipativity

To guarantee robust stability in the presence of arbitrary bounded inputs, basic passivity is not enough. We need to ensure the system's internal dynamics are uniformly contractive, that is, they dissipate energy at a guaranteed rate, regardless of the operating mode selected by the *selection signal* $x(\cdot)$. This ensures the system can actively counteract the energy being injected by the *port input* $u(\cdot)$.

With this subsection, we present such a condition based on the existence of a common quadratic Lyapunov function demonstrating uniform dissipativity.

**Theorem 7.1** (Global Stability from Uniform Local Dissipativity)**.** Consider the continuous-time selective state-space model equation 2 under the standard regularity assumptions. Assume further that:

(i) There exists a time-varying quadratic Lyapunov function candidate $V_Q(t, h) = \frac{1}{2}h^H Q(t)h$, where $Q : \mathbb{T} \to \mathbf{S}_+^N(\mathbb{C})$ satisfies:

  - $Q \in W_{\text{loc}}^{1,1}(\mathbb{T}, \mathbf{S}_+^N(\mathbb{C}))$ (absolutely continuous locally).
  - $Q(t)$ is uniformly positive definite and bounded: there exist constants $k_2 \geq k_1 > 0$ such that $k_1 I \preceq Q(t) \preceq k_2 I$ for all $t \in \mathbb{T}$.

(ii) The dynamics satisfy a uniform dissipativity condition with respect to $V_Q$: There exists a constant $\delta > 0$ such that for almost every $t \in \mathbb{T}$ and for **all** admissible *selection* values $x \in \mathbb{C}^{d_{\text{in}}}$ that can occur at time $t$, the following matrix inequality holds:

$$\dot{Q}(t) + Q(t)A\big(\Delta(t), x\big) + A\big(\Delta(t), x\big)^H Q(t) \preceq -2\delta Q(t) \tag{20}$$

(This implies that for any fixed selection value $x$, the homogeneous LTV system $\dot{h} = A(\Delta(t), x)h$ satisfies $\dot{V}_Q \leq -2\delta V_Q$ and hence is uniformly exponentially stable; equivalently, $\|h(t)\| \leq \sqrt{k_2/k_1}\, e^{-\delta(t-t_0)}\|h(t_0)\|$.)

(iii) The input matrix $B(\Delta(t), x)$ is uniformly bounded: There exists a constant $M_B > 0$ such that $\|B(\Delta(t), x)\|_2 \leq M_B$ for all $t$ and admissible $x$.

Then, the selective state-space system equation 2 is globally ISS with respect to the *port input* $u(t)$, uniformly over all admissible selection schedules $x(\cdot)$. Specifically, there exist constants $\tilde{C} \geq 1$, $\tilde{\gamma} > 0$, and a class $\mathcal{K}$ gain function $\sigma$ such that for any initial state $h(t_0)$, any admissible selection schedule $x(\cdot)$, and any admissible port input $u(\cdot)$, the solution satisfies:

$$\|h(t)\| \leq \tilde{C}e^{-\tilde{\gamma}(t-t_0)}\|h(t_0)\| + \sigma\left(\|u\|_{[t_0,t],\infty}\right), \qquad \|u\|_{[t_0,t],\infty} := \operatorname*{ess\,sup}_{t_0 \leq \tau \leq t} \|u(\tau)\|. \quad \text{for all } t \geq t_0. \tag{21}$$

*Proof Sketch.* We analyze the time derivative of the Lyapunov function $V_Q$ along the system trajectories. The derivative splits into two parts: one from the internal dynamics ($A$ matrix) and one from the external *port input* ($B$ matrix). The uniform dissipativity assumption provides a strong negative bound on the internal part ($\leq -2\delta V_Q$). The input part is bounded using the Cauchy-Schwarz inequality. Combining these results in a differential inequality of the form $\frac{dV}{dt} \leq -aV + b\sqrt{V}$. By a change of variables, $\Psi = \sqrt{V}$, this transforms into a standard linear differential inequality, $\frac{d\Psi}{dt} \leq -c\Psi + d$, whose solution is bounded. Translating this bound back to the state norm $\|h(t)\|$ yields the classic Input-to-State Stability (ISS) estimate. Check Appendix A.7 for the full proof. □

**ML implication.** *Theorem 7.1 is the practically useful version of the structural results in Section 5. Where Theorem 5.2 characterized what passivity would have to enforce on a selective SSM under a single quadratic certificate, the ISS condition equation 20 is what robustness actually requires: a uniform exponential decay rate of the homogeneous dynamics, in matrix-inequality form, that holds across every selection value gating can produce. The discrete LMI regularizer of Section 6 can be read as a soft enforcement of exactly this kind of uniform-in-x matrix inequality on the Mamba selective-scan core: the regularizer drives $\lambda_{\max}(\mathcal{L}_k(P, \beta))$ toward zero across all tokens, which is the sampled analogue of asking equation 20 to hold for every selection value at almost every time. The empirical question this raises — does enforcing the discrete analog actually improve the model's behavior under driving inputs and perturbations? — is what Section 8 is designed to answer.*

**Remark 7.2** (Relationship to the Passivity LMI)**.** The ISS condition in Eq. equation 20 is conceptually stronger than the passivity LMI from Theorem 5.2. The passivity LMI describes an **energy balance**, ensuring the system does not generate energy. In contrast, the ISS condition demands a forced **energy decay**, ensuring the system's internal dynamics actively dissipate energy at a uniform exponential rate.

Formally, if the ISS condition holds (with $\delta > 0$ and a positive definite $Q$), it is sufficient to satisfy the internal stability portion (the (1,1) block) of the passivity LMI. However, a system can be passive (energy-balanced) without being uniformly contractive in this stricter sense. Thus, the ISS condition is a specialized and powerful tool for proving robust stability, while the passivity LMI is a more general tool for analyzing a system's energy flow. In Section 8, we therefore use the more permissive passivity LMI (in its sampled form from Section 6) as the training-time regularizer, and treat the stronger ISS-style condition as the gold standard that the regularizer is empirically pushing the Mamba core toward.

## 8 Empirical Validation on SST/Mamba

The continuous-time analysis of Sections 4–7 and the discrete bridge of Section 6 make a concrete prediction: penalizing the largest positive eigenvalue of the sampled block matrix $\mathcal{L}_k(P, \beta)$ from equation 18 during training should constrain the Mamba selective-scan core toward sampled passivity, ideally without harming downstream forecasting accuracy. This section empirically evaluates that prediction across a real Mamba-based architecture on seven standard time-series forecasting datasets and four prediction horizons. Our goal

is not to set new MSE records on these benchmarks; it is to ask, with honest scoping, whether the discrete LMI regularizer in equation 19 actually does what the theory says it should, and what limits of the framework the experiments expose. Extended per-dataset tables, training hyperparameters, and additional ablations are deferred to Appendix C.

### 8.1 Experimental Setup

**Architecture.** We use the SST forecasting architecture (Xu et al., 2025), which is a routed mixture of a Mamba long-range expert and a short-range attention branch. We apply our regularizer only to the Mamba branch: at each token $k$, the per-step matrices $(F_k, G_k, C_k, D_k)$ are extracted from the Mamba selective-scan core during the forward pass, the block matrix $\mathcal{L}_k(P, \beta)$ from equation 18 is formed, and the regularizer equation 19 is added to the standard task loss. Importantly, this means the regularizer is a Mamba-core diagnostic rather than a passivity certificate for the full SST network (with normalization, residual connections, and routing); we discuss this scoping carefully throughout.

**Datasets and horizons.** We evaluate on seven standard forecasting datasets: ETTh1, ETTh2, ETTm1, ETTm2, weather, electricity, and traffic, with prediction horizons $\{96, 192, 336, 720\}$, giving $7 \times 4 = 28$ dataset–horizon pairs. The look-back window is 672 tokens; the label length is 336.

**Variants.** We compare:

- **SST baseline.** The unregularized SST architecture.

- **SST + discrete LMI ($P = I$).** SST trained with $\mathcal{R}_{\mathrm{LMI}}$ at $P = I$, $\lambda_{\mathrm{LMI}} = 10^{-2}$, $\beta = 10^{-3}$. This is our main practical method.

- **SST + discrete LMI (diag-$Q$).** As above with $P = \mathrm{diag}(q)$ a learnable diagonal storage (initialized at $q = \mathbf{1}$). Reported as a conservativeness ablation.

Additional ablations (a milder $\lambda_{\mathrm{LMI}}$ regime, a projection step, and a router-energy variant) appear in Sections 8.4–8.5 and Appendix C.

**Diagnostics.** At evaluation time we record clean test MSE and MAE, plus four Mamba-core diagnostics: the mean sampled LMI violation $\mathbb{E}_k[\max(0, \lambda_{\max}(\mathcal{L}_k))]$, the maximum sampled $\lambda_{\max}(\mathcal{L}_k)$, the mean Mamba-core SSM state norm, and (when relevant) the router's mean Mamba weight $m_{\mathrm{weight}}$.

### 8.2 Discrete LMI Regularization Reduces Mamba-Core Passivity Violations at No MSE Cost

We first evaluate, on clean test data, whether enforcing $\mathcal{L}_k(P, \beta) \preceq 0$ during training actually shifts the Mamba core toward sampled passivity. Table 3 summarizes the aggregate result across all 28 dataset–horizon pairs.

| Variant | Clean MSE | Clean MAE | Mean LMI viol. | Max $\lambda_{\max}(\mathcal{L}_k)$ |
|---|---|---|---|---|
| SST baseline | 0.385804 | 0.390236 | 0.035824 | 2.6012 |
| SST + discrete LMI ($P = I$) | 0.385815 | 0.390244 | **0.002882** | **1.8972** |
| SST + discrete LMI (diag-$Q$) | 0.385815 | 0.390243 | 0.002877 | 1.8975 |

Table 3: Aggregate clean-test results across 28 dataset–horizon pairs, mean over runs. The discrete LMI regularizer reduces the sampled Mamba-core LMI violation by approximately 92% relative to baseline, while clean MSE differs by less than 0.003%.

The aggregate picture is unambiguous. Mean Mamba-core LMI violation drops from 0.0358 to 0.0029 under $P = I$ regularization, an approximately 92% reduction; the largest positive eigenvalue $\lambda_{\max}(\mathcal{L}_k)$ across all tokens and runs drops from 2.6012 to 1.8972. Clean MSE, by contrast, changes by approximately 0.0012% on

average, with a maximum per-run delta of 0.018%. Crucially, the violation reduction holds in 28/28 paired dataset–horizon comparisons, with the largest absolute reductions on the datasets where the unregularized baseline violates the criterion most (traffic, electricity), and the smallest on datasets where the baseline already nearly satisfies it (the ETTh family). Figure 1 shows this per-dataset structure.

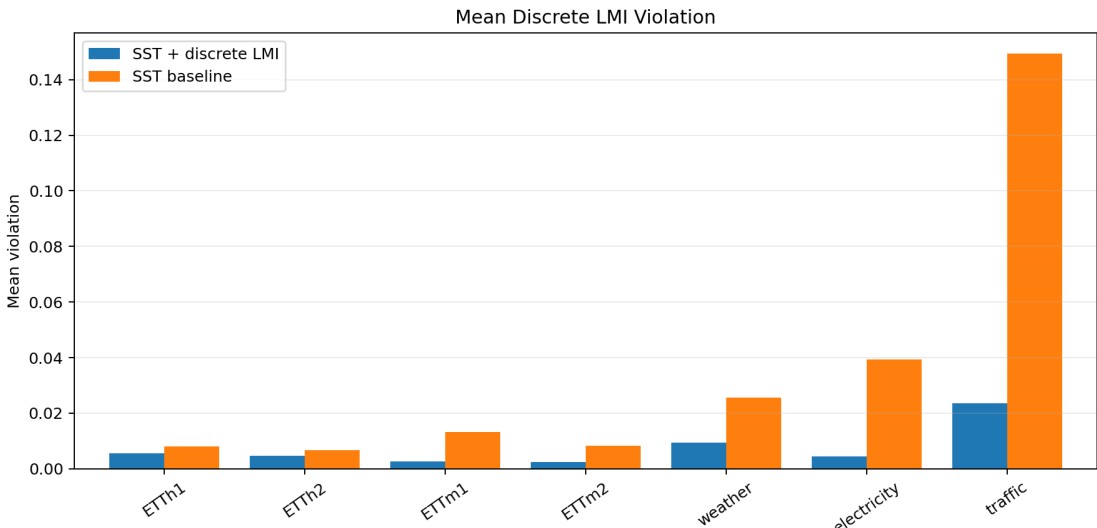

Figure 1: Mean sampled Mamba-core LMI violation per dataset, averaged across the four prediction horizons. The discrete LMI regularizer reduces the violation on every dataset, with the largest reductions exactly where the unregularized baseline violates the sampled criterion most strongly.

We read this as the cleanest empirical statement of what Section 6 predicted: the regularizer does what it is designed to do, and the cost in forecasting accuracy is in the noise. Two notes on scoping. First, although $\lambda_{\max}(\mathcal{L}_k)$ is reduced uniformly, it is not driven below zero; the regularizer produces a substantial and consistent *reduction in violation*, not a certified passivity statement. Second, this is a Mamba-core property; the surrounding SST network (normalization, residuals, routing) is not constrained by the regularizer.

### 8.3 Conservativeness Ablation: $P = I$ vs Learnable Diagonal $P$

A natural reviewer concern is that $P = I$ might be unduly conservative as a storage choice. Section 6 explicitly noted that any $P \succeq 0$ is admissible, so we test whether replacing $P = I$ with a learnable diagonal $P = \mathrm{diag}(q)$ buys anything. Aggregate results are in Table 4.

| Variant | Clean MSE | Mean LMI viol. | $q_{\mathrm{mean}}$ | $q_{\mathrm{log\text{-}std}}$ |
|---|---|---|---|---|
| SST + discrete LMI ($P = I$) | 0.385815 | 0.002882 | – | – |
| SST + discrete LMI (diag-$Q$) | 0.385815 | 0.002877 | 1.0008 | $1.1 \times 10^{-3}$ |

Table 4: Conservativeness ablation: replacing the storage $P = I$ with a learnable diagonal $P = \mathrm{diag}(q)$ produces a strict (if small) Pareto improvement on 28/28 pairs at no measurable accuracy cost.

Across all 28 paired runs, the diagonal-$Q$ variant has lower mean LMI violation than $P = I$, with no MSE degradation (paired-run violation reductions range from $-0.10\%$ to $-0.33\%$; MSE deltas in the noise floor at $\sim 10^{-5}$). Per-run reductions are strict: every paired run sits in the favourable corner of the (violation reduction, MSE delta) plane. The learned $q$ moves visibly away from identity ($q_{\mathrm{mean}} \approx 1.0008$, with $q_{\mathrm{log\text{-}std}} \approx 1.1 \times 10^{-3}$ and $q_{\max} \approx 1.0084$ on the dataset where it pushes hardest, traffic), confirming that the diagonal storage is being used. The improvement is modest in absolute terms; we read this as evidence that the framework can support less-conservative storage metrics without harming accuracy, and we use $P = I$ as the default practical choice while reporting diag-$Q$ as the cleaner conservativeness check.

### 8.4 Robustness Diagnostics under Injected Perturbations

So far, our empirical claims concern clean test inputs. We next ask whether the regularizer changes the Mamba core's behavior when inputs are corrupted. We use five synthetic perturbation modes applied only to the test inputs (target sequences are kept clean): Gaussian noise, spike injections, block masks, amplitude shifts, and last-value-hold; each variant is evaluated three times across seeds. Aggregate results, averaged across the five non-clean modes and all 28 dataset–horizon pairs, are in Table 5.

| Variant | Robust MSE | MSE deg. % | Mean LMI viol. | Mean max state norm |
|---|---|---|---|---|
| SST baseline | 0.70808 | +126.76% | 0.02345 | 0.26651 |
| SST + discrete LMI ($P = I$) | 0.70809 | +126.76% | **0.00512** | **0.25588** |
| SST + discrete LMI (diag-$Q$) | 0.70809 | +126.76% | 0.00512 | 0.25588 |

Table 5: Robustness diagnostics under five injected perturbation modes (Gaussian noise, spikes, block masks, amplitude shifts, last-value-hold), averaged across 28 dataset–horizon pairs and 3 seeds. The regularizer reduces internal Mamba-core LMI violations and state norms uniformly, but does not change robust forecasting MSE.

The result is sharply two-sided, and we report it as such. On internal Mamba-core diagnostics under perturbation, the regularizer is a strict Pareto improvement: mean LMI violation drops from 0.02345 to 0.00512 ($\sim 60\%$ reduction; lower in 35/35 dataset–mode pairs), mean max SSM state norm from 0.26651 to 0.25588 ($\sim 3.4\%$, lower in 35/35 pairs), and the worst observed state norm in the experiment from approximately 0.6065 to 0.5387. These improvements hold uniformly across all five perturbation modes, with the largest LMI-violation reductions on the hardest perturbation (amplitude shift, $\sim 345\%$ MSE degradation against clean for the baseline). On forecasting MSE under perturbation, by contrast, the regularizer produces no measurable improvement: robust MSE deltas vs. baseline are at +0.0008%, well within numerical noise. The Pareto picture under amplitude shift, the most challenging perturbation, is shown in Figure 2.

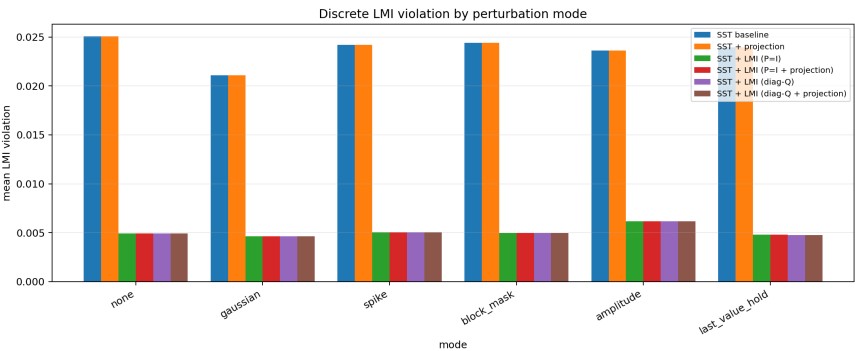

Figure 2: Pareto plot under amplitude perturbation, aggregated across datasets. Each point is one variant. The four LMI-regularized variants cluster strictly inside the favorable corner (lower mean Mamba-core LMI violation *and* lower mean max SSM state norm) of the (state norm, LMI violation) plane, while baseline and projection-only fall in the unfavorable corner.

We are intentionally precise about how we frame this. The regularizer improves internal Mamba stability and passivity diagnostics under perturbed inputs without degrading forecasting accuracy. It does not, in this run, produce a robust-MSE improvement; this is an honest negative on the practical task metric. Two plausible reasons are worth flagging. First, the regularizer is a Mamba-core diagnostic; the surrounding SST network (normalization, residual paths, routing, and output heads) absorbs much of the perturbation effect on MSE, so improving the Mamba core internally does not by itself translate into improved end-to-end MSE under corruption. Second, the perturbations are injected at test time only, so there is no opportunity for the regularized model to align its representation with the perturbed regime. We discuss both in Section 8.6.

## 8.5 Router-Stability Ablation: A Neutral Result

The SST router decides, per sample, how much to weight the Mamba branch versus the short-range branch. A natural conjecture is that adding an energy-aware penalty on the router — one that discourages high Mamba weight on samples with high local LMI violation — could induce a learned "avoid the Mamba branch when the Mamba core is locally unstable" rule, of the kind a human reading Section 7 would write. We tested this with a small additional regularizer on the router during training. Aggregate results over the same 28 dataset–horizon pairs are in Table 6.

| Variant | Clean MSE | Mean $m_w$ | Train Corr$(m_w, \text{viol.})$ | Mean LMI viol. |
|---|---|---|---|---|
| SST + discrete LMI ($P = I$) | 0.385815 | 0.30801 | – | 0.00754 |
| SST + discrete LMI + router | 0.385815 | 0.30799 | +0.0043 | 0.00754 |

Table 6: Router-stability ablation: adding the energy-aware router penalty preserves clean MSE in 28/28 runs and consistently nudges the Mamba weight downward, but the train-time correlation between Mamba weight and LMI violation is essentially zero on aggregate, with inconsistent sign across datasets ($-0.107$ on ETTh1 to $+0.054$ on ETTm1). We report this as a clean-performance neutrality result rather than as evidence of a learned energy-aware routing schedule. We use $m_w := m_{\text{weight}}$ for brevity.

The result is honest neutral. Clean MSE is preserved exactly within tolerance (28/28 within 1%), and the Mamba weight is lower than the LMI-only variant in 28/28 runs — directionally aligned with the energy-aware reading. But the magnitude of the Mamba-weight shift is on the order of $10^{-5}$, the train-time correlation between Mamba weight and local LMI violation is $+0.0043$ on aggregate (essentially zero), and per-dataset correlations span both signs (from $-0.107$ on ETTh1 to $+0.054$ on ETTm1). We do not interpret this as evidence of a learned energy-aware schedule, and we present it as an appendix-style ablation rather than a positive contribution. The most plausible reason the mechanism is inert is that the router is trained on clean data, where local LMI violation is not a strong predictor of task loss. We expect that retraining with injected perturbations, so that violation magnitude becomes correlated with task loss, could produce a more consistent energy-aware schedule, but we leave this as future work.

## 8.6 Limitations

Three limitations of the empirical evaluation deserve explicit statement, and they should also be read as the limits of what the present paper claims.

**Mamba core, not full network.** The discrete LMI regularizer equation 19 is a per-token, per-channel constraint on the Mamba selective-scan recurrence equation 16. Sections 8.2–8.4 consequently report Mamba-core diagnostics, not passivity certificates for the full SST network. The mismatch between strong improvements in Mamba-core diagnostics and the absence of robust-MSE improvement (Section 8.4) is the empirical face of this scoping. Bridging this gap would require either a network-level passivity criterion (a research direction the parametric LMI of Section 5 suggests but does not yet provide) or an explicit cascade argument.

**Sampled criterion, not exact discrete passivity.** Proposition 6.1 gives an exact one-step dissipativity inequality at the Mamba selective-scan core; what we enforce during training is a soft penalty on the largest positive eigenvalue. Empirically, $\lambda_{\max}(\mathcal{L}_k)$ is reduced uniformly but not driven below zero. The numerical results should therefore be read as "substantial reduction in violation" rather than "certified sampled passivity". We do not claim the latter.

**Single architecture family.** All experiments are on SST as the host architecture, a routed Mamba/short-range mixture. We chose this because it gives a clean Mamba branch on which to evaluate the criterion, and because the router lets us also probe the energy-aware routing question of Section 8.5. The criterion of Section 6 is, however, generic: it depends only on the per-token matrices $(F_k, G_k, C_k, D_k)$ extracted from any selective-scan core. Validation on a wider set of selective architectures (HGRN, GLA, vanilla Mamba without the short-range branch, hybrid selective Transformers) is left as future work.

In summary, the discrete LMI regularizer is empirically clean-MSE safe, substantially reduces sampled Mamba-core LMI violations on every clean and perturbed run we tested, and improves Mamba-core stability diagnostics under input corruption. It does not, on its own, produce a robust-MSE improvement, and we explicitly do not claim it does. Section 9 discusses how these scoped empirical results sharpen the theoretical claims of Sections 4–7 and what they imply for selective-architecture design.

## 9 Conclusion

This paper has done two things. On the theoretical side, we developed a passivity- and ISS-based account of continuous-time selective State-Space Models with discontinuous gating, paying particular attention to the structural regularity of energy storage under the kind of $L^p_{\text{loc}}$ coefficient functions that gating actually produces. On the practical side, we translated that account into a sampled, differentiable criterion on the Mamba selective-scan core and evaluated it as a training-time regularizer on a real Mamba-based forecasting architecture across 28 dataset–horizon pairs. We close by pulling together what was proved, what was empirically supported, and what was deliberately not claimed.

By separating the selection signal $x(\cdot)$ from the driving (port) input $u(\cdot)$, we obtained four structural results for system equation 2. State-strict dissipativity, together with quadratic bounds on a storage functional, yields exponential forgetting of homogeneous trajectories uniformly over admissible selection schedules (Theorem 4.1). Freezing the selection produces a passive linear time-varying input–output subsystem whose minimal available storage is necessarily quadratic with $\text{AUC}_{\text{loc}}$ regularity (Theorem 4.2), giving a canonical intrinsic energy structure that is robust to discontinuous gating. Under the strong but informative hypothesis that a single quadratic storage certifies passivity uniformly over admissible selection schedules, we derived a parametric LMI in $x$ together with universal kernel/unobservability and rank-monotonicity conditions, formalizing what we called *irreversible forgetting* (Theorems 5.1–5.4). Finally, a uniform-in-$x$ matrix inequality on the homogeneous dynamics yields global ISS with respect to the port input (Theorem 7.1).

Section 6 translated the parametric LMI of Theorem 5.2 into a one-step dissipativity inequality on the Mamba selective-scan recurrence equation 16, with a sampled block matrix $\mathcal{L}_k(P, \beta)$ whose structure mirrors the continuous-time LMI exactly (Proposition 6.1). The largest positive eigenvalue of this matrix is differentiable in the model parameters that the Mamba selective scan produces, so it can be used directly as a soft training-time constraint equation 19. We are explicit that this criterion is a property of the Mamba core, not a passivity certificate for the surrounding deep network.

On clean test data across seven standard time-series datasets and four horizons, the discrete LMI regularizer reduces mean Mamba-core LMI violation by approximately 92% relative to the unregularized baseline (28/28 pairs), while clean MSE differs by less than 0.018% (Section 8.2). Replacing $P = I$ with a learnable diagonal $P = \text{diag}(q)$ produces a strict (if small) Pareto improvement in 28/28 paired runs at no measurable accuracy cost (Section 8.3), confirming that the framework is not overly tied to the simplest storage choice. Under five injected input perturbations, the regularizer produces a uniform $\sim 60\%$ reduction in mean Mamba-core LMI violation and a $\sim 3.4\%$ reduction in mean max SSM state norm, holding in 35/35 dataset–mode pairs and on every perturbation type (Section 8.4).

We have been deliberately precise about what the empirics do not say. Under perturbed inputs, the regularizer improves Mamba-core stability and passivity diagnostics but does not produce a robust forecasting MSE improvement on its own (Section 8.4); we report this as an honest negative on the practical task metric. The sampled LMI is reduced but not certified negative semidefinite, so we say "substantial reduction in violation" rather than "certified sampled passivity". The router-stability ablation preserves clean MSE and consistently nudges Mamba weights downward but does not produce a learned energy-aware routing schedule, with train-time correlations between Mamba weight and LMI violation that are essentially zero on aggregate and inconsistent in sign across datasets (Section 8.5). The universal-storage hypothesis used in Section 5 is a strong sufficient condition that motivates the practical regularizer; it is not a claim that every Mamba in the wild satisfies it.

Read together, the theory and the experiments suggest three concrete design lessons for selective SSMs. First, principled initializations such as HiPPO (Gu et al., 2020) are best understood as the design of a

baseline energy landscape on which input-driven selection then operates: Theorem 4.2 explains why such initializations admit a well-conditioned quadratic energy function, and Appendix B.1 shows empirically that initializations failing this admit no valid energy function at all and diverge. Second, the sampled criterion of Section 6 is a low-cost, differentiable Mamba-core diagnostic that can be added to existing training pipelines with negligible overhead and provably (in the empirical sense of 28/28 paired runs) reduces sampled passivity violations without harming accuracy: it is, at minimum, a useful structural regularizer even if its end-to-end MSE benefit is not a forced consequence of passivity. Third, the structural framework of Sections 4–5 gives a precise control-theoretic vocabulary — forgetting, kernel inertness, irreversibility — in which to phrase questions about expressivity–stability tradeoffs in selective architectures, going beyond the Lyapunov-exponent diagnostics that have so far dominated the literature on Mamba-style stability (Halloran et al., 2024).

The honest scoping in Section 8.6 identifies three concrete next steps. (i) *From Mamba core to deep network.* The most informative gap revealed by our experiments is between Mamba-core diagnostic improvements and absent end-to-end MSE improvements under perturbation. Closing this gap requires either a deep-network passivity criterion (a research direction the parametric LMI of Theorem 5.2 suggests but does not yet resolve) or an explicit cascade argument bridging core dissipativity and end-to-end robustness. (ii) *Perturbation-aware energy routing.* The router-stability ablation suggests, plainly, that an energy-aware routing schedule does not emerge from training on clean inputs, because local LMI violation is not a strong predictor of clean-data task loss. We expect that retraining with injected perturbations, so that violation magnitude becomes correlated with task loss, is a path to a genuine learned routing rule of the kind a reader of Section 7 would write. (iii) *Discrete-time theorems.* The continuous-time results of Sections 4–7 have direct discrete-time analogues whose proofs we have left informal in Section 6; making the rank-monotonicity and irreversibility statements rigorous in the sampled regime, and connecting them to the empirically observed monotonic-reduction behaviour in Section 8, is the natural rigorous extension of the present work.

Selective SSMs are mathematically awkward: they are neither LTI, nor classical LTV, nor smooth nonlinear systems. We have argued that, because they are nonetheless port-Hamiltonian-style systems with a clear supply rate, they are amenable to a passivity-based analysis whose continuous-time conclusions transfer cleanly to the discrete Mamba selective scan as a per-token training-time criterion. Within the boundaries we have drawn, the criterion does what the theory predicts and reveals where the theory currently stops. We hope this combination of structural results and honest empirical scoping is a useful step toward a control-theoretic foundation for selective architectures in deep learning, and toward concrete tools that practitioners can drop into existing training loops without paying for them in accuracy.

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
