# A  Theoretical Proofs

## A.1  Proof of Theorem 4.1

We are given the existence of a storage function $V(t, h)$ satisfying the strict dissipativity inequality equation 5 and the quadratic bounds equation 6. The regularity assumption allows us to consider the differential version of the inequality. Along any trajectory of the system equation 2, the time derivative of $V$ satisfies:

$$\frac{d}{dt}V\big(t, h(t)\big) \leq \text{Re} \langle u(t), y(t) \rangle - \beta \|h(t)\|^2 \tag{22}$$

almost everywhere. This follows from differentiating the integral inequality or directly from the definition of dissipativity via differential supply rates. Now, consider the homogeneous (zero port-input) case, where $u(t) \equiv 0$ for all $t \geq 0$. Then the supply term vanishes:

$$\text{Re} \langle u(t), y(t) \rangle = \text{Re} \langle 0, y(t) \rangle = 0. \tag{23}$$

(As a special case, under frozen selection $x(t) \equiv 0$ this homogeneous trajectory coincides with setting $u \equiv 0$ in equation 10, i.e., $\dot{h} = A_0(t)h$ and $y_0 = C_0(t)h$.)

Substituting $u(t) = 0$ into the differential inequality equation 22, we get:

$$\frac{d}{dt}V\big(t, h(t)\big) \leq -\beta \|h(t)\|^2 \tag{24}$$

for trajectories in the homogeneous (zero port-input) case.

We can relate $\|h(t)\|^2$ back to $V(t, h(t))$ using the upper quadratic bound from equation 6: $\|h(t)\|^2 \geq \frac{1}{k_2}V(t, h(t))$. Substituting this into equation 24:

$$\frac{d}{dt}V\big(t, h(t)\big) \leq -\frac{\beta}{k_2}V\big(t, h(t)\big). \tag{25}$$

Let $\Phi(t) = V(t, h(t))$. This is a scalar non-negative function satisfying the differential inequality $\dot{\Phi}(t) \leq -\gamma'\Phi(t)$, where $\gamma' = \beta/k_2 > 0$. By the Comparison Lemma (a consequence of Grönwall's inequality), this implies:

$$\Phi(t) = V\big(t, h(t)\big) \leq V\big(0, h(0)\big) e^{-\gamma' t} \quad \text{for all } t \geq 0. \tag{26}$$

Now, we use both quadratic bounds from equation 6:

- Lower bound on $V(t, h(t))$: $k_1 \|h(t)\|^2 \leq V(t, h(t))$.

- Upper bound on $V(0, h(0))$: $V(0, h(0)) \leq k_2 \|h(0)\|^2$.

Combining these with the decay inequality equation 26:

$$k_1 \|h(t)\|^2 \leq V\big(t, h(t)\big) \leq V\big(0, h(0)\big) e^{-\gamma' t} \leq k_2 \|h(0)\|^2 e^{-\gamma' t}. \tag{27}$$

Dividing by $k_1 > 0$:

$$\|h(t)\|^2 \leq \frac{k_2}{k_1} \|h(0)\|^2 e^{-\gamma' t}. \tag{28}$$

Taking the square root of both sides:

$$\|h(t)\| \leq \sqrt{\frac{k_2}{k_1}} \|h(0)\| e^{-(\gamma'/2)t}. \tag{29}$$

Defining $C = \sqrt{k_2/k_1} \geq 1$ (since $k_2 \geq k_1 > 0$) and $\gamma = \gamma'/2 = \beta/(2k_2) > 0$, we obtain the desired exponential decay:

$$\|h(t)\| \leq C e^{-\gamma t} \|h(0)\|. \tag{30}$$

This completes the proof.

*Summary:* This theorem establishes that intrinsic energy dissipation ($\beta > 0$), when coupled with a storage function $V$ that is quadratically comparable to the state norm, guarantees exponential stability of the homogeneous system ($u \equiv 0$), and in particular of the frozen-selection homogeneous dynamics ($u \equiv 0$, $x \equiv 0$). This signifies a fundamental "forgetting" capability, ensuring that the influence of the initial state diminishes exponentially over time in the absence of external port input, regardless of the specific (potentially non-quadratic) nature of $V$.

### A.2 Available storage for passive LTV systems

We use the notion of *available storage* for LTV systems as defined in Morandin & Hinsen (2024, Def. 4.1): for an LTV system of the form $\dot{h} = A(t)h + B(t)u$, $y = C(t)h + D(t)u$, the available storage is

$$V_a(t_0, h_0) = \sup_{\substack{t_1 \geq t_0,\, u \in L^2_{\text{loc}} \\ h(t_0) = h_0}} \left( -\int_{t_0}^{t_1} \text{Re} \left\langle u(t), y(t) \right\rangle \, dt \right).$$

For passive LTV systems (in the sense of Morandin & Hinsen (2024, Def. 1.1)), the available storage is finite and coincides with a *minimal quadratic* storage $V_a = V_Q$ for some $Q \in \text{AUC}_{\text{loc}}$ (Morandin & Hinsen, 2024, Thm. 4.2, Cor. 4.3).

### A.3 Proof of Theorem 4.2

We prove the three items (a)–(c).

**(a) Passivity of the frozen-selection LTV system.** Fix the selection signal $x(t) \equiv 0$ and define

$$A_0(t) := A(\Delta(t), 0), \qquad B_0(t) := B(\Delta(t), 0), \qquad C_0(t) := C(\Delta(t), 0), \qquad D_0(t) \equiv 0.$$

By the regularity assumptions of the main text, $A_0 \in L^1_{\text{loc}}$, $B_0, C_0 \in L^2_{\text{loc}}$, and $D_0 \in L^\infty_{\text{loc}}$, hence the LTV system

$$\dot{h}(t) = A_0(t)h(t) + B_0(t)u(t), \qquad y_0(t) = C_0(t)h(t) + D_0(t)u(t)$$

is of the form Morandin & Hinsen (2024, Eq. (1)). Let $(h(\cdot), u(\cdot), y_0(\cdot))$ be any state–input–output solution of this system on $[t_0, T]$. It is also a trajectory of the full selective SSM equation 2 when restricting to $x(\cdot) \equiv 0$. Therefore, applying equation 9 yields

$$V(T, h(T)) - V(t_0, h(t_0)) \leq \int_{t_0}^{T} \text{Re} \left\langle u(\tau), y_0(\tau) \right\rangle \, d\tau - \beta \int_{t_0}^{T} \|h(\tau)\|^2 \, d\tau \leq \int_{t_0}^{T} \text{Re} \left\langle u(\tau), y_0(\tau) \right\rangle \, d\tau.$$

Thus $V$ satisfies the passivity dissipation inequality of Morandin & Hinsen (2024, Def. 1.1) for the frozen-selection LTV system, i.e., the LTV system is passive.

**(b) Existence of minimal quadratic available storage.** Let $V_{a,0}$ be the available storage of the frozen-selection LTV system in the sense of Morandin & Hinsen (2024, Def. 4.1). Since the system is passive, Morandin & Hinsen (2024, Cor. 4.3) implies that $V_{a,0}$ is finite and coincides with a quadratic form:

$$V_{a,0}(t, h) = \tfrac{1}{2} h^H Q_0(t) h$$

for some $Q_0 : \mathbb{T} \to \mathbf{S}^N_+(\mathbb{C})$, and that $V_{a,0}$ is minimal among all storage functions.

**(c) $\text{AUC}_{\text{loc}}$ regularity of $Q_0(t)$.** The same result Morandin & Hinsen (2024, Cor. 4.3) yields that $Q_0$ can be chosen such that $Q_0 \in \text{AUC}_{\text{loc}}(\mathbb{T}, \mathbf{S}^N_+(\mathbb{C}))$.

**Connection to exponential decay.** The homogeneous (unforced) trajectories correspond to setting $u \equiv 0$ in the frozen-selection LTV system, i.e., $\dot{h} = A_0(t)h$ (Morandin & Hinsen, 2024, Sec. 2.1). If $V$ furthermore satisfies the hypotheses of Theorem 4.1, then Theorem 4.1 gives exponential decay along these homogeneous trajectories.

## A.4 Proof of Theorem 5.1

(a) and (b): The premise is that $V_Q(t, h)$ satisfies the passivity inequality equation 11 for all admissible trajectories of the selective SSM equation 2. As argued in the proof of Theorem 4.2(a), any such trajectory includes the trajectories of the frozen-selection LTV system equation 10 obtained by setting $x(t) \equiv 0$. Therefore, $V_Q$ must also be a valid storage function for the frozen-selection LTV system equation 10: indeed, since equation 11 holds for all admissible pairs $(x(\cdot), u(\cdot))$, it holds in particular for $x(\cdot) \equiv 0$ and for every admissible $u(\cdot)$, yielding the dissipation inequality for all trajectories of equation 10. Dropping the nonpositive term $-\beta \int_{t_0}^{T} \|h(\tau)\|^2 \, d\tau$ yields the passivity inequality of Morandin & Hinsen (2024, Def. 1.1).

The framework of Morandin & Hinsen (Morandin & Hinsen, 2024) analyzes quadratic storage functions for passive LTV systems under the assumed $L_{\text{loc}}^p$ regularity. Specifically, (Morandin & Hinsen, 2024, Theorem 3.2) establishes necessary conditions for a quadratic form $V_Q(t, h) = \frac{1}{2} h^H Q(t) h$ to be a storage function. It implies that $Q$ must be absolutely upper semi-continuous (which corresponds to $\text{AUC}_{\text{loc}}$ in their terminology). By Morandin & Hinsen (2024, Thm. 3.2), any quadratic storage function $V_Q(t, h) = \frac{1}{2} h^H Q(t) h$ for a passive LTV system must be induced by a matrix function $Q \in \text{AUC}_{\text{loc}}(\mathbb{T}, \mathbf{S}_+^N(\mathbb{C}))$. Then Morandin & Hinsen (2024, Thm. 5.4(1)) yields that $t \mapsto \text{rank}(Q(t))$ is weakly decreasing on $\mathbb{T}$.

*Summary:* This theorem reveals that the strong requirement of universal passivity guaranteed by a single $V_Q$ immediately imposes significant structure on $Q(t)$ itself. It must possess $\text{AUC}_{\text{loc}}$ regularity, accommodating potential discontinuities but ensuring they behave in a controlled manner (e.g., jumps cannot increase energy in the Loewner sense). Crucially, the rank monotonicity implies that the dimension of the subspace captured by the energy function (its image) cannot increase over time. This lays the groundwork for understanding irreversible effects, as the "energy-less" subspace (the kernel) can only grow or stay the same.

## A.5 Proof of Theorem 5.2

(a) **Parametric LMI Condition:** The dissipativity inequality equation 12 holding for all trajectories implies the following a.e. differential inequality along trajectories (at times where $\dot{Q}(t)$ exists): $\frac{d}{dt} V_Q(t, h(t)) \leq \text{Re} \langle u(t), y(t) \rangle - \beta \|h(t)\|^2$. Substituting $V_Q = \frac{1}{2} h^H Q h$, $\dot{h} = A(t, x)h + B(t, x)u$, and $y = C(t, x)h$ leads to a quadratic inequality in $(h, u)$, with the selection value $x$ acting as a parameter:

$$\frac{1}{2} h^H \dot{Q} \, h + \text{Re}(h^H Q \dot{h}) \leq \text{Re}(u^H C(t, x)h) - \beta \|h\|^2. \tag{31}$$

$$\frac{1}{2} h^H \dot{Q} \, h + \text{Re}(h^H Q(A(t, x)h + B(t, x)u)) \leq \text{Re}(u^H C(t, x)h) - \beta \|h\|^2. \tag{32}$$

Rearranging terms:

$$\frac{1}{2} h^H \dot{Q} \, h + \text{Re}(h^H Q A(t, x)h) + \text{Re}(h^H Q B(t, x)u) - \text{Re}((C(t, x)^H u)^H h) + \beta \, h^H h \leq 0. \tag{33}$$

$$\frac{1}{2} h^H (\dot{Q} + Q A(t, x) + A(t, x)^H Q + 2\beta I) h + \text{Re}(h^H (Q B(t, x) - C(t, x)^H)u) \leq 0. \tag{34}$$

For each fixed admissible selection value $x$ and for almost every time $t$, this inequality must hold for all $h \in \mathbb{C}^N$ and all port inputs $u \in \mathbb{C}^{d_{\text{in}}}$. This is precisely the condition encoded by the negative semidefiniteness of the LMI matrix $\mathcal{L}(t, x)$ defined in equation 13. Since $V_Q$ must work for any input trajectory, the LMI must hold parametrically for all input values $x$ that can occur at time $t$. This leads directly to the LMI formulation (setting $D = 0$):

$$\begin{bmatrix} \dot{Q}(t) + Q(t)A(t, x) + A(t, x)^H Q(t) + 2\beta I & Q(t)B(t, x) - C(t, x)^H \\ B(t, x)^H Q(t) - C(t, x) & 0 \end{bmatrix} \preceq 0 \tag{35}$$

(b) **Universal Kernel Condition for** $C$**:** Let $v \in \text{Ker}(Q(t))$ at a time $t$ where the LMI equation 13 holds. Consider the augmented vector $z = \begin{bmatrix} v \\ w \end{bmatrix}$ for any $w \in \mathbb{C}^{d_{\text{in}}}$. The LMI implies $z^H \mathcal{L}(t, x) z \leq 0$.

$$z^H \mathcal{L}(t, x) z = \begin{bmatrix} v^H & w^H \end{bmatrix} \begin{bmatrix} \dot{Q} + QA + A^H Q + 2\beta I & QB - C^H \\ B^H Q - C & 0 \end{bmatrix} \begin{bmatrix} v \\ w \end{bmatrix}$$
$$= v^H (\dot{Q} + QA + A^H Q + 2\beta I) v + v^H (QB - C^H) w + w^H (B^H Q - C) v + w^H (0) w$$

Since $v \in \text{Ker}(Q(t))$, we have $Q(t)v = 0$ and $v^H Q(t) = 0$. The expression simplifies to:

$$= v^H (\dot{Q} + 0 + 0 + 2\beta I) v + v^H (0 - C^H) w + w^H (0 - C) v$$
$$= v^H \dot{Q} v + 2\beta \|v\|^2 - v^H C^H w - w^H C v$$
$$= v^H \dot{Q} v + 2\beta \|v\|^2 - 2 \text{Re}(w^H C v)$$

This quadratic form in $w$ must be $\leq 0$. Considering the $2 \times 2$ projection onto $v$ and $w$ space: $\begin{bmatrix} v^H (\dot{Q} + QA + A^H Q + 2\beta I) v & v^H (QB - C^H) w \\ w^H (B^H Q - C) v & 0 \end{bmatrix} = \begin{bmatrix} v^H \dot{Q} v + 2\beta \|v\|^2 & -v^H C(t, x)^H w \\ -w^H C(t, x) v & 0 \end{bmatrix} \preceq 0$. For this $2 \times 2$ matrix to be negative semidefinite, the diagonal elements must be non-positive (taking $w = 0$ shows $v^H (\dot{Q} + QA + A^H Q + 2\beta I) v \leq 0$, and if $v \in \text{Ker } Q(t)$ this reduces to $v^H \dot{Q}(t) v + 2\beta \|v\|^2 \leq 0$), and the determinant must be non-negative. The determinant is $(v^H \dot{Q} v + 2\beta \|v\|^2)(0) - |-w^H C(t, x) v|^2 = -|w^H C(t, x) v|^2$. So we need $-|w^H C(t, x) v|^2 \geq 0$. This can only hold if $|w^H C(t, x) v|^2 = 0$ for all $w \in \mathbb{C}^{d_{\text{in}}}$. This implies $C(t, x)v = 0$. Since this must hold for the $C(t, x)$ corresponding to any admissible $x$ at time $t$, we conclude $\forall v \in \text{Ker}(Q(t))$, $C(\Delta(t), x)v = 0$ for all admissible $x$, a.e. $t$.

(c) **Implicit Constraints on** $A$ **and** $B$**:** The validity of the LMI equation 13 for all $x$ directly imposes constraints on $A(t, x)$ via the $(1, 1)$ block and couples $B(t, x)$ to $C(t, x)$ (which is already constrained by part (b)) via the off-diagonal blocks and $Q(t)$. These ensure that the dynamics generated by any input $x$ remain compatible with the energy storage/dissipation defined by $V_Q$.

*Summary:* This theorem translates the abstract requirement of universal passivity into a concrete parametric LMI equation 13. This LMI must hold not just for a specific input or for the unforced system, but simultaneously for all possible selection values $x$ that the gating mechanism might encounter at any given time $t$. A striking consequence is the universal kernel condition equation 14: any state direction $v$ that is considered "energy-less" by the storage function (i.e., $v \in \text{Ker}(Q(t))$) must be rendered unobservable at the output ($C(t, x)v = 0$), irrespective of the specific input $x$ driving the gating. This imposes a strong limitation on the gating mechanism: it cannot arbitrarily change the output matrix $C$ in response to input $x$ in a way that would make these energy-less states visible. Passivity demands consistency between the energy accounting ($Q$) and observability ($C$).

## A.6 Proof of Theorem 5.4

1. **Kernel Non-Shrinking:** This follows directly from Theorem 5.1(b), which established that $\text{rank}(Q(t))$ is weakly monotonically non-increasing. Since $\dim(\text{Ker}(Q(t))) = N - \text{rank}(Q(t))$, a non-increasing rank implies a non-decreasing kernel dimension. Therefore $t \mapsto \dim \text{Ker}(Q(t))$ is weakly non-decreasing.

2. **Energy Consistency within the Kernel:** This condition was derived within the proof of Theorem 5.2(b) by evaluating the $(1, 1)$ block of the parametric LMI equation 13 for a vector $h(t) \in \text{Ker}(Q(t))$. The $(1, 1)$ block inequality is $\dot{Q}(t) + Q(t)A(t, x) + A(t, x)^H Q(t) + 2\beta I \preceq 0$. Pre- and post-multiplying by $h(t)^H$ and $h(t)$ respectively, and using $Q(t)h(t) = 0$, yields $h(t)^H \dot{Q}(t) h(t) + 2\beta \|h(t)\|^2 \leq 0$. This inequality must hold independently of $x$ because it follows from the LMI which holds parametrically.

3. **Constraint on Dynamics Violating Kernel Structure:** If dynamics induced by some $x^*(t)$ were fundamentally incompatible with the passivity condition guaranteed by $V_Q$ (e.g., by attempting to move a state out of the kernel in an "energy-creating" way relative to $V_Q$), it would necessarily cause the parametric LMI equation 13 to fail for $x = x^*(t)$. This contradicts the core assumption that $V_Q$ ensures passivity

universally. Therefore, all admissible dynamics under the universal $V_Q$ assumption must inherently respect the energy balance constraints, including those within the kernel.

*Summary:* This theorem formalizes the notion of "irreversible forgetting" within the framework of a universal quadratic storage function. The non-shrinking kernel (Property 1) implies that once a state direction is deemed irrelevant or forgotten from an energy perspective (i.e., enters $\mathrm{Ker}(Q(t))$), it structurally remains so according to the fixed energy measure $Q(t)$. The gating mechanism cannot manipulate $Q(t)$ to "un-forget" this mode. Furthermore, Property 2 ensures that the system dynamics and the evolution of $Q(t)$ itself must maintain energy consistency within this forgotten subspace, respecting the required dissipation rate $\beta$. Any gating strategy $x(t)$ must induce dynamics $(A(t,x), B(t,x), C(t,x))$ compatible with these constraints (Property 3). This provides a lens for analyzing robust memory properties: modes in $\mathrm{Ker}(Q(t))$ are stably forgotten. It also offers potential insights into phenomena like catastrophic forgetting in learning systems; if learning adapts a $Q(t)$-like structure, changes that violate kernel inertness or energy consistency might be necessary to learn new conflicting information, potentially disrupting the established passive structure.

### A.7 Proof of Theorem 7.1

We analyze the time derivative of the Lyapunov function candidate $V_Q(t, h(t)) = \frac{1}{2}h(t)^H Q(t) h(t)$ along the trajectories of the full system equation 2. Since $Q \in W^{1,1}_{\mathrm{loc}}$, its derivative $\dot{Q}(t)$ exists a.e. and lies in $L^1_{\mathrm{loc}}$. Using the chain rule and substituting the dynamics $\dot{h}(t) = A(\Delta(t), x(t))h(t) + B(\Delta(t), x(t))u(t)$, we obtain

$$
\begin{aligned}
\frac{d}{dt}V_Q(t, h(t)) &= \frac{1}{2}h^H \dot{Q}h + \frac{1}{2}\dot{h}^H Q h + \frac{1}{2}h^H Q \dot{h} \\
&= \frac{1}{2}h^H \dot{Q}h + \mathrm{Re}\big(h^H Q \dot{h}\big) \\
&= \frac{1}{2}h^H \dot{Q}h + \mathrm{Re}\big(h^H Q \left(A(t,x(t))h + B(t,x(t))u(t)\right)\big) \\
&= \underbrace{\frac{1}{2}h^H \left(\dot{Q}(t) + Q(t)A(t,x(t)) + A(t,x(t))^H Q(t)\right)h}_{\text{Homogeneous Part}} + \underbrace{\mathrm{Re}\big(h^H Q(t)\, B(t,x(t))\, u(t)\big)}_{\text{Input Part}}.
\end{aligned}
$$

Here, $A(t, x(t))$ stands for $A(\Delta(t), x(t))$, and similarly for $B$.

**1. Homogeneous part.** By the uniform dissipativity assumption equation 20, which holds for the specific selection value $x = x(t)$ occurring at time $t$, we have

$$
\frac{1}{2}h^H \left(\dot{Q}(t) + Q(t)A(t,x(t)) + A(t,x(t))^H Q(t)\right)h \leq \frac{1}{2}h^H(-2\delta Q(t))h = -\delta h^H Q(t)h = -2\delta V_Q(t, h(t)).
$$

**2. Input part.** We bound the input term using $|\Re(z)| \leq |z|$ and Cauchy–Schwarz:

$$
\begin{aligned}
\left|\mathrm{Re}\big(h^H Q(t)B(t,x(t))u(t)\big)\right| &\leq \left|h^H Q(t)B(t,x(t))u(t)\right| \\
&\leq \|h(t)\| \, \|Q(t)\|_2 \, \|B(t,x(t))\|_2 \, \|u(t)\| \\
&\leq \|h(t)\| \, k_2 \, M_B \, \|u(t)\| \qquad (\text{since } Q(t) \preceq k_2 I \text{ and } \|B(\Delta(t),x)\|_2 \leq M_B).
\end{aligned}
$$

Using $Q(t) \succeq k_1 I$, we have

$$
V_Q(t, h(t)) = \frac{1}{2}h^H Q(t)h \geq \frac{k_1}{2}\|h(t)\|^2 \quad \Rightarrow \quad \|h(t)\| \leq \sqrt{\frac{2}{k_1}}\,\sqrt{V_Q(t, h(t))}.
$$

Therefore,

$$
\left|\mathrm{Re}\big(h^H Q(t)B(t,x(t))u(t)\big)\right| \leq \left(\sqrt{\frac{2}{k_1}}\,k_2 M_B\right)\|u(t)\|\sqrt{V_Q(t, h(t))}. \tag{36}
$$

**3. Differential inequality for $V_Q$.** Combining the homogeneous and input bounds yields, for almost every $t$,

$$\frac{d}{dt}V_Q(t, h(t)) \leq -2\delta V_Q(t, h(t)) + \left(\sqrt{\frac{2}{k_1}}\, k_2 M_B\right) \|u(t)\| \sqrt{V_Q(t, h(t))}. \tag{37}$$

Let $\Phi(t) = V_Q(t, h(t)) \geq 0$ and define $K := \sqrt{\frac{2}{k_1}}\, k_2 M_B$. Then

$$\dot{\Phi}(t) \leq -2\delta\Phi(t) + K\|u(t)\|\sqrt{\Phi(t)} \qquad \text{for a.e. } t. \tag{38}$$

**4. Comparison inequality via $\Psi = \sqrt{\Phi}$.** Define $\Psi(t) = \sqrt{\Phi(t)}$. For almost every $t$ such that $\Phi(t) > 0$,

$$\dot{\Psi}(t) = \frac{\dot{\Phi}(t)}{2\sqrt{\Phi(t)}} \leq \frac{-2\delta\Phi(t) + K\|u(t)\|\sqrt{\Phi(t)}}{2\sqrt{\Phi(t)}} = -\delta\Psi(t) + \frac{K}{2}\|u(t)\|.$$

Hence,

$$\dot{\Psi}(t) \leq -\delta\Psi(t) + \frac{K}{2}\|u(t)\| \qquad \text{for a.e. } t. \tag{39}$$

By the comparison principle (integrating factor argument),

$$\Psi(t) \leq e^{-\delta(t-t_0)}\Psi(t_0) + \int_{t_0}^{t} e^{-\delta(t-\tau)}\frac{K}{2}\|u(\tau)\|\, d\tau. \tag{40}$$

Let

$$\|u\|_{[t_0, t], \infty} := \operatorname*{ess\,sup}_{t_0 \leq \tau \leq t} \|u(\tau)\|.$$

Then

$$\begin{aligned}
\int_{t_0}^{t} e^{-\delta(t-\tau)}\frac{K}{2}\|u(\tau)\|\, d\tau &\leq \frac{K}{2}\|u\|_{[t_0, t], \infty} \int_{t_0}^{t} e^{-\delta(t-\tau)}\, d\tau \\
&= \frac{K}{2}\|u\|_{[t_0, t], \infty} \cdot \frac{1 - e^{-\delta(t-t_0)}}{\delta} \\
&\leq \frac{K}{2\delta}\|u\|_{[t_0, t], \infty}.
\end{aligned}$$

Therefore,

$$\Psi(t) \leq e^{-\delta(t-t_0)}\Psi(t_0) + \frac{K}{2\delta}\|u\|_{[t_0, t], \infty}. \tag{41}$$

**5. Convert back to $\|h(t)\|$.** Substituting $\Psi = \sqrt{V_Q}$ and using $k_1 I \preceq Q(t) \preceq k_2 I$ (for all $t$),

$$\sqrt{V_Q(t, h(t))} \geq \sqrt{\frac{k_1}{2}}\, \|h(t)\|, \qquad \sqrt{V_Q(t_0, h(t_0))} \leq \sqrt{\frac{k_2}{2}}\, \|h(t_0)\|.$$

Thus,

$$\begin{aligned}
\sqrt{\frac{k_1}{2}}\, \|h(t)\| &\leq \sqrt{V_Q(t, h(t))} \\
&\leq e^{-\delta(t-t_0)}\sqrt{V_Q(t_0, h(t_0))} + \frac{K}{2\delta}\|u\|_{[t_0, t], \infty} \\
&\leq e^{-\delta(t-t_0)}\sqrt{\frac{k_2}{2}}\, \|h(t_0)\| + \frac{K}{2\delta}\|u\|_{[t_0, t], \infty}.
\end{aligned}$$

Dividing by $\sqrt{k_1/2}$ gives

$$\|h(t)\| \leq \sqrt{\frac{k_2}{k_1}}\, e^{-\delta(t-t_0)}\|h(t_0)\| + \underbrace{\frac{k_2 M_B}{\delta k_1}}_{=:K'}\|u\|_{[t_0, t], \infty}. \tag{42}$$

This is exactly the ISS estimate equation 21 with $\tilde{C} = \sqrt{k_2/k_1}$, $\tilde{\gamma} = \delta$, and $\sigma(s) = K's$.

*Summary:* Under a common quadratic Lyapunov function $V_Q$ with uniform bounds and a uniform decay rate equation 20 holding for all admissible selection values $x$ (hence for any selection schedule $x(\cdot)$), the selective SSM is ISS with respect to the port input $u(\cdot)$, with a linear gain depending on $k_1, k_2, \delta$ and the uniform bound $M_B$ on $B(\Delta(t), x)$.

### A.8 Explanation of Comparison Principle and derivation of Eq. (40)

We start from the differential inequality derived for $\Psi(t)$:

$$\dot{\Psi}(t) \leq -\delta\Psi(t) + \frac{K}{2}\|u(t)\| \tag{43}$$

which can be rewritten as

$$\dot{\Psi}(t) + \delta\Psi(t) \leq \frac{K}{2}\|u(t)\| \tag{44}$$

where $\delta > 0$. This is a standard first-order linear differential inequality. Its solution bound can be obtained either by the explicit method of integrating factors or by invoking a suitable comparison lemma. Both approaches lead to the same upper bound, namely Eq. equation 40.

**Method 1: Integrating factor (standard ODE technique).** Multiply both sides of equation 44 by the integrating factor $e^{\delta t}$:

$$e^{\delta t}\dot{\Psi}(t) + \delta e^{\delta t}\Psi(t) \leq e^{\delta t}\frac{K}{2}\|u(t)\|. \tag{45}$$

Recognize the left-hand side as the derivative of a product:

$$\frac{d}{dt}\big(e^{\delta t}\Psi(t)\big) = e^{\delta t}\dot{\Psi}(t) + \delta e^{\delta t}\Psi(t). \tag{46}$$

Hence,

$$\frac{d}{dt}\big(e^{\delta t}\Psi(t)\big) \leq e^{\delta t}\frac{K}{2}\|u(t)\|. \tag{47}$$

Integrating both sides from $t_0$ to $t$ (using $\tau$ as the integration variable) gives

$$\int_{t_0}^{t} \frac{d}{d\tau}\big(e^{\delta\tau}\Psi(\tau)\big)\, d\tau \leq \int_{t_0}^{t} e^{\delta\tau}\frac{K}{2}\|u(\tau)\|\, d\tau. \tag{48}$$

By the Fundamental Theorem of Calculus,

$$e^{\delta t}\Psi(t) - e^{\delta t_0}\Psi(t_0) \leq \int_{t_0}^{t} e^{\delta\tau}\frac{K}{2}\|u(\tau)\|\, d\tau. \tag{49}$$

Solving for $\Psi(t)$ yields

$$\begin{aligned}
\Psi(t) &\leq e^{-\delta(t-t_0)}\Psi(t_0) + e^{-\delta t}\int_{t_0}^{t} e^{\delta\tau}\frac{K}{2}\|u(\tau)\|\, d\tau \\
&= e^{-\delta(t-t_0)}\Psi(t_0) + \int_{t_0}^{t} e^{-\delta(t-\tau)}\frac{K}{2}\|u(\tau)\|\, d\tau,
\end{aligned} \tag{50}$$

which is exactly Eq. equation 40.

**Method 2: Comparison lemma (a Grönwall-type argument).** A common comparison lemma states the following: if $\eta(\cdot)$ is absolutely continuous and satisfies

$$\dot{\eta}(t) \leq a(t)\eta(t) + b(t) \quad \text{a.e. on } [t_0, t],$$

and if $\zeta(\cdot)$ solves the corresponding equality

$$\dot{\zeta}(t) = a(t)\zeta(t) + b(t), \qquad \zeta(t_0) = \eta(t_0),$$

then $\eta(t) \leq \zeta(t)$ for all $t \geq t_0$.

In our case, equation 43 has $a(t) = -\delta$ and $b(t) = \frac{K}{2}\|u(t)\|$. The solution of the equality $\dot{\zeta} = -\delta\zeta + \frac{K}{2}\|u(t)\|$ with $\zeta(t_0) = \Psi(t_0)$ is given by variation of constants:

$$\zeta(t) = e^{-\delta(t-t_0)}\Psi(t_0) + \int_{t_0}^{t} e^{-\delta(t-\tau)}\frac{K}{2}\|u(\tau)\|\,d\tau. \tag{51}$$

By the comparison lemma, $\Psi(t) \leq \zeta(t)$, which again yields Eq. equation 40.

## B  Toy Simulations Illustrating the Continuous-Time Theory

This appendix presents three small simulation studies that illustrate the structural results of Sections 4–5. The runs here are deliberately small-scale: they use synthetic dynamics rather than real Mamba selective scans, and their purpose is pedagogical, not benchmark-setting. The full empirical evaluation of the discrete LMI regularizer on a real Mamba-based forecasting architecture appears in Section 8, with extended experimental details in Appendix C.

### B.1  Initialization's Decisive Impact on the Energy Landscape

**Objective.** This experiment provides a concrete demonstration of the core claims in Section 4: that a model's initialization directly governs its intrinsic stability and memory-vs-stability trade-off. It illustrates Theorem 4.2 by exhibiting cases where a quadratic energy function $V_{a,0}(h) = \frac{1}{2}h^T Q_0 h$ exists and behaves as the theory predicts, and a control case where it does not.

**Setup.** We compared three initialization strategies for the frozen-selection state matrix $A_0(t) = A(\Delta(t), 0)$ of an 8-dimensional SSM ($N = 8$).

1. **HiPPO-style initialization ($A_H$):** Eigenvalues are generated to be stable (negative real parts) but clustered near the imaginary axis, with the rightmost (slowest) eigenvalue at $\mathrm{Re}(\lambda) \approx -0.1$.

2. **Random Stable initialization ($A_S$):** Eigenvalues are randomly generated but constrained to be in the left half-plane, resulting in a wider spectral spread. The rightmost eigenvalue is at $\mathrm{Re}(\lambda) \approx -0.52$.

3. **Random Unstable initialization ($A_U$):** Eigenvalues are drawn from the same distribution as $A_S$ but without the stability constraint, resulting in a dominant eigenvalue at $\mathrm{Re}(\lambda) \approx +0.79$.

For each $A_0$, we first attempted to find a corresponding quadratic energy matrix $Q_0$ by solving the continuous-time Lyapunov equation $A_0^T Q_0 + Q_0 A_0 = -I$. We then simulated the frozen-selection system ($x(t) \equiv 0$) by feeding it a white-noise *port input* $u(t)$ for 5 seconds to populate its state, after which the port input was turned off ($u(t) \equiv 0$) to observe the free energy decay.

**Results & Interpretation.** Our findings, summarized in Figure 3, provide strong empirical validation for our theory:

- **Existence of a Valid Energy Function ($Q_0$):** Both the *HiPPO-style* and *Random Stable* initializations admitted a unique, positive-definite solution $Q_0$ to the Lyapunov equation (with condition numbers of $\approx 21$ and $\approx 13$, respectively). The *Random Unstable* initialization failed to produce a valid solution, yielding an indefinite matrix, confirming that no quadratic energy function exists for an unstable system.

- **Spectral Gap Governs Memory:** The slowest decay rate is dictated by the rightmost eigen-value. The spectral gap for HiPPO ($\approx -0.1\,s^{-1}$) was five times smaller than for Random Stable ($\approx -0.52\,s^{-1}$). Consequently, after the input was removed, the state norm of the HiPPO-initialized model took $\approx 23$ seconds to decay by two orders of magnitude, while the Random Stable model did so in just $\approx 4$ seconds. The Unstable model diverged exponentially.

- **Energy Trajectories Confirm Theory:** The energy $V_{a,0}(t) = \frac{1}{2}h(t)^T Q_0 h(t)$ for the HiPPO model decayed almost perfectly linearly on a log-scale with a slope of $\approx -0.1$, illustrating a long memory horizon. The Random Stable model's energy decayed five times faster.

This experiment confirms that merely being stable is insufficient. The precise spectral placement of $A_0$, as achieved by principled initializations like HiPPO, directly governs the memory-vs-stability trade-off, exactly as our energy-based theoretical framework predicts.

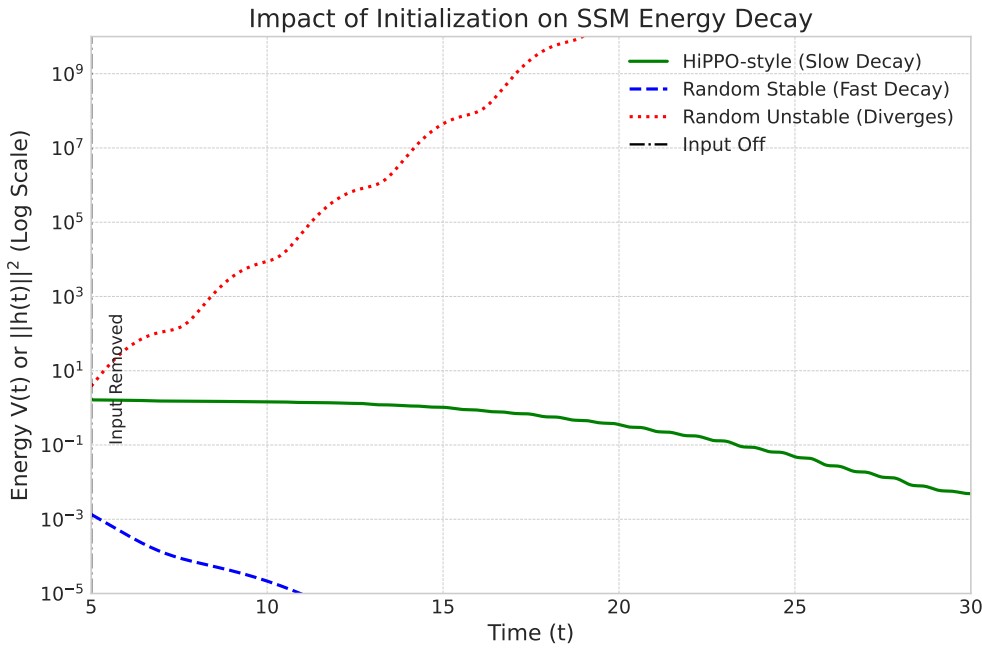

Figure 3: Energy decay $V_{a,0}(t)$ on a log-scale for an 8-D SSM with three different initializations after the port input $u(t)$ is removed (set to zero) at $t = 5$s. The HiPPO-style initialization yields a valid energy function and a slow decay rate (long memory). The Random Stable initialization also has a valid energy function but decays much faster (short memory). The Unstable initialization has no valid energy function and diverges.

## B.2    Visualizing Irreversible Forgetting via Rank-Deficient Gating

**Objective.**  This experiment is designed to provide a concrete, visual demonstration of the "irreversible forgetting" concept introduced in Section 5. It illustrates Theorem 5.1 (rank monotonicity of any universal quadratic storage matrix) and Theorem 5.4 (kernel inertness) on a small synthetic switched system.

**Setup.**  We construct a 3-dimensional linear system $\dot{h} = A(t)h$ that can be switched between two operating modes via a gating signal. Each mode is defined by a state matrix $A_i$ and an associated minimal storage matrix $Q_i$. The storage matrices are the unique positive semidefinite solutions to the continuous-time Lyapunov equation $A_i^T Q_i + Q_i A_i = -C_i^T C_i$, which defines the energy landscape for an observable system. Modes of the system are:

- **Mode 1 (Full-Rank Storage):**  The system dynamics are governed by $A_1 = \mathbf{diag}[-0.2, -0.3, -0.4]$. We choose $C_1 = I_3$, representing full observability of the state. The re-

sulting storage matrix $Q_1$ is full-rank (rank=3), meaning the system can store energy in all three state dimensions.

- **Mode 2 (Rank-Deficient Storage):** The dynamics are governed by $A_2 = \mathbf{diag}[-0.2, -0.3, -15]$. The large negative eigenvalue is designed to rapidly dissipate the third state component. We choose $C_2 = \mathbf{diag}[1, 1, 0]$, making the third state dimension unobservable. The resulting storage matrix $Q_2$ is rank-deficient (rank=2), with the third dimension lying in its kernel ($\mathrm{Ker}(Q_2)$). In this mode, the system loses the capacity to store energy in the third dimension.

**Simulation Protocol.** We simulate the homogeneous system (no input channel) from a random initial state with non-zero components in all dimensions. The gating signal switches the system's mode according to the following timeline:

- **0s $\leq t <$ 5s:** The system operates in Mode 1 (full-rank energy storage).

- **5s $\leq t <$ 10s:** The gating switches the system to Mode 2 (rank-deficient storage).

- $t \geq$ **10s:** The gating attempts to switch the system back to Mode 1.

**Theoretical Prediction & Results.** Our theory predicts that if a single, universal quadratic storage function $V_Q(t)$ governs the entire trajectory, its defining matrix $Q(t)$ must have a non-increasing rank. More precisely, in this case:

- The switch at $t = 5$s is permissible, as the storage rank can decrease from 3 to 2.

- The attempted switch back at $t = 10$s, however, would necessitate an increase in the storage rank from 2 to 3. This violates the rank monotonicity property inherent to $\mathrm{AUC_{loc}}$ functions.

This implies that no single $V_Q(t)$ can certify passivity for the entire trajectory. The "forgetting" of the energy storage capacity in the third dimension is, from the perspective of a single passive structure, irreversible.

The simulation results, visualized in Figure 4, confirm this prediction empirically. During Mode 2, the third state component ($h_3$) rapidly collapses to zero and, crucially, does not recover even after the system dynamics are switched back to Mode 1. The system becomes permanently confined to the two-dimensional subspace. This provides a clear, empirical illustration of our theoretical claims: once the system's energy storage rank collapses due to gating, it cannot be increased again without violating the fundamental constraints required for a stable, passive system.

### B.3 LMI Regularization for Improved Training Robustness on a Toy System

**Objective.** This experiment validates, on a small synthetic system, the central practical claim that motivated the discrete-time bridge of Section 6: that the continuous-time LMI condition from Theorem 5.2 can be used as a regularizer to train more robust selective SSMs. The full evaluation of the discrete-time analogue of this regularizer on a real Mamba-based architecture is in Section 8.

**Setup.**

- **Task:** We designed a challenging 1D tracking task where the SSM's output $y(t)$ must track a "spiky" reference signal $r(t)$ composed of steps and sharp impulses, designed to provoke large internal state excursions.

- **Model:** We use a 2-dimensional ($N = 2$) selective SSM. We use the coupled case $u(t) \equiv x(t)$, so the same signal both selects parameters and drives the dynamics. The state matrix is explicitly selection-dependent to ensure selectivity: $A(x) = A_{\mathrm{base}} + \tanh(x) \cdot A_{\mathrm{sel}}$, where $A_{\mathrm{base}}$ and $A_{\mathrm{sel}}$ are learned $2 \times 2$ matrices. The input matrix $B$ and output matrix $C$ are also learned.

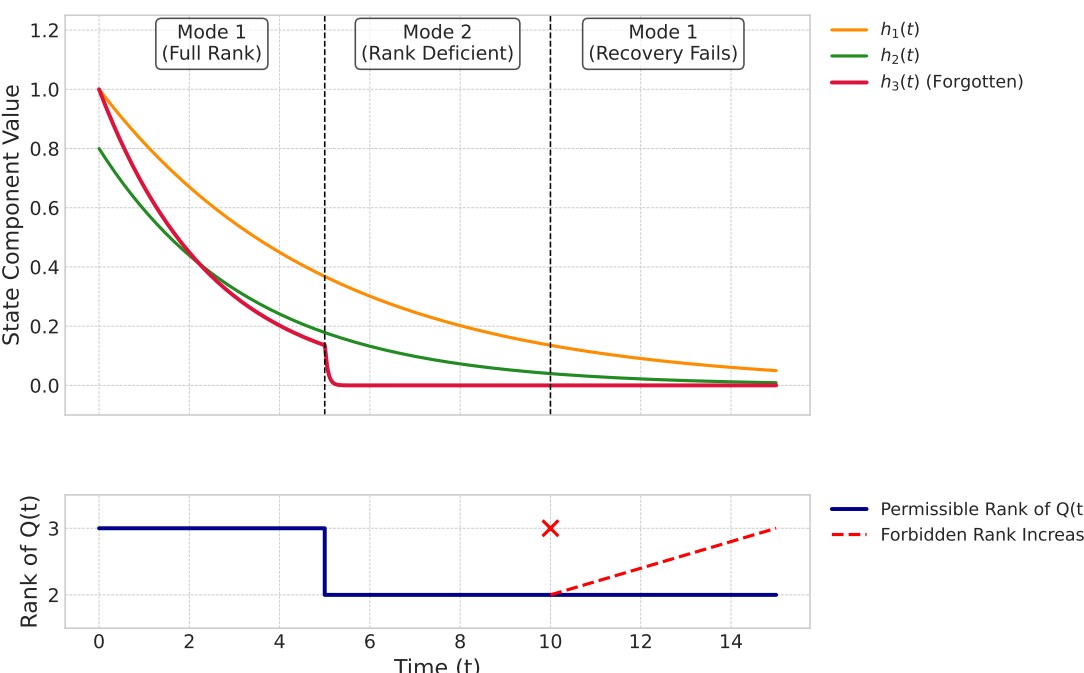

Figure 4: Illustration of irreversible forgetting. The rank of the underlying minimal storage matrix is plotted over time (or implied by state trajectories). The system switches from a full-rank mode (rank=3) to a rank-deficient mode (rank=2) at $t = 5$s, causing the third state component to collapse. Our theory proves that any attempt to switch back in a way that increases the rank of a universal storage function (e.g., the dashed red line) is incompatible with maintaining a single, passive energy structure, demonstrating the irreversibility principle from Theorem 5.1.

- **LMI Regularizer:** We employ the LMI from Theorem 5.2. For simplicity and to demonstrate the core principle, we fix the storage matrix $Q = I$. The regularization loss is defined as the magnitude of the LMI violation: $\mathcal{L}_{\text{LMI}} = \max(0, \lambda_{\max}(\mathcal{L}(x)))$, where $\mathcal{L}(x)$ is the LMI matrix evaluated for the selection value $x$ (equivalently for the port input since $u \equiv x$ in this experiment).

- **Two Training Conditions:** We compare two models: 1. **Baseline Model:** Trained only with the task loss, $\mathcal{L}_{\text{total}} = \mathcal{L}_{\text{task}}$, where $\mathcal{L}_{\text{task}} = \text{MSE}(y(t), r(t))$. 2. **LMI-Regularized Model:** Trained with the combined loss, $\mathcal{L}_{\text{total}} = \mathcal{L}_{\text{task}} + \gamma \cdot \mathcal{L}_{\text{LMI}}$, using a small regularization weight $\gamma = 0.01$.

**Results and Interpretation.** The results of the experiment, evaluated on a held-out test set, are summarized in Table 7.

Table 7: Comparison of Baseline vs. LMI-Regularized SSM on a challenging tracking task.

| Model | Task MSE (Test) | Max State Norm $\|h\|_\infty$ | Max LMI Violation |
|---|---|---|---|
| Baseline Model | 0.073 | 18.4 | 3.51 |
| LMI-Regularized Model | 0.081 | **1.9** | **0.003** |

The results lead to three clear conclusions:

1. **Robustness is Dramatically Improved:** The most interesting result is in the maximum state norm. The baseline model, while achieving a slightly better task MSE, does so by allowing its

internal state to reach a very large norm (18.4). This is indicative of a model operating on the edge of instability, vulnerable to exploding states. In contrast, the **LMI-regularized model's state norm is an order of magnitude smaller (1.9)**. It has learned to perform the task while keeping its internal dynamics constrained and provably more stable.

2. **The Regularizer Works as Intended:** The "Max LMI Violation" column shows that the regularizer was highly effective. The baseline model frequently and significantly violates the passivity condition (max violation of 3.51). The regularized model has learned parameters that keep the LMI matrix negative semidefinite (max violation is near zero), successfully enforcing the theoretical condition for stability derived in our work.

3. **Minimal Impact on Task Performance:** Crucially, this significant gain in robustness comes at a negligible cost to task performance. The MSE of the regularized model is only marginally higher than the baseline, demonstrating that a stable, well-behaved solution exists that is also effective for the task.

To conclude, this experiment demonstrates on a small toy system that our theoretical condition can be directly translated into a practical tool for improving training robustness. The discrete-time analogue of this regularizer, applied to a real Mamba selective-scan core, is evaluated systematically in Section 8.

## C   Extended Mamba/SST Experimental Details

This appendix collects the experimental details that did not fit in Section 8. We give the full training and architecture configuration in Appendix C.1, additional per-dataset and per-run breakdowns of the headline discrete LMI result in Appendix C.2, the milder LMI-weight regime that complements the conservativeness ablation in Appendix C.3, the full per-mode breakdown of the robustness study in Appendix C.4, and the per-dataset breakdown of the router-stability ablation in Appendix C.5. Aggregate analysis CSVs and additional plots are released alongside the paper.

### C.1   Architecture, Datasets, and Training Configuration

**Architecture.**   We use SST (Xu et al., 2025) as the host architecture. SST routes a Mamba long-range expert and a short-range branch via a learned router; we apply our discrete LMI regularizer only to the Mamba branch. For the smaller datasets (ETTh1, ETTh2, ETTm1, ETTm2, weather), Mamba is configured with model dimension $d_m = 16$, two heads, $d_f = 64$. For the larger datasets (electricity, traffic), $d_m = 32$, four heads, $d_f = 128$. All other architectural details follow the SST defaults.

**Datasets and horizons.**   We use the seven public time-series benchmarks ETTh1, ETTh2, ETTm1, ETTm2, weather, electricity, and traffic, with the standard train/val/test splits. The look-back window is seq_len = 672, the label length is label_len = 336, and the prediction horizons are pred_len $\in \{96, 192, 336, 720\}$, giving the $7 \times 4 = 28$ dataset–horizon pairs reported throughout Section 8.

**Training.**   All variants use the SST default optimizer, learning-rate schedule, and number of epochs. The discrete LMI regularizer of equation 19 is added to the standard task loss with weight $\lambda_{\mathrm{LMI}} \in \{10^{-3}, 10^{-2}\}$ depending on the regime: $\lambda_{\mathrm{LMI}} = 10^{-3}$ is the milder weight studied in Appendix C.3; $\lambda_{\mathrm{LMI}} = 10^{-2}$ is the headline weight used in Sections 8.2–8.3, with $\beta = 10^{-3}$ and $P \in \{I, \mathrm{diag}(q)\}$. The optional projection step that re-PSDifies $P$ after each update was effectively non-binding in our runs (yielding outputs visually identical to the non-projection variant on every diagnostic), so we report results for it only briefly in Appendix C.4 and do not present it as a central component. Each variant is trained with three seeds; the aggregate numbers in Section 8 are means over seeds.

### C.2   Discrete LMI Regularization: Per-Run Breakdown

The aggregate result of Section 8.2 is that the discrete LMI regularizer reduces mean Mamba-core LMI violation by approximately 92% (28/28 pairs) while clean MSE differs by less than 0.018% per run. The

per-run picture, in the form of paired MSE deltas vs. the unregularized baseline, is shown in Figure 5. The per-dataset breakdown of how strongly $\lambda_{\max}(\mathcal{L}_k)$ is reduced is shown in Figure 6, complementing the mean-violation plot in the main paper.

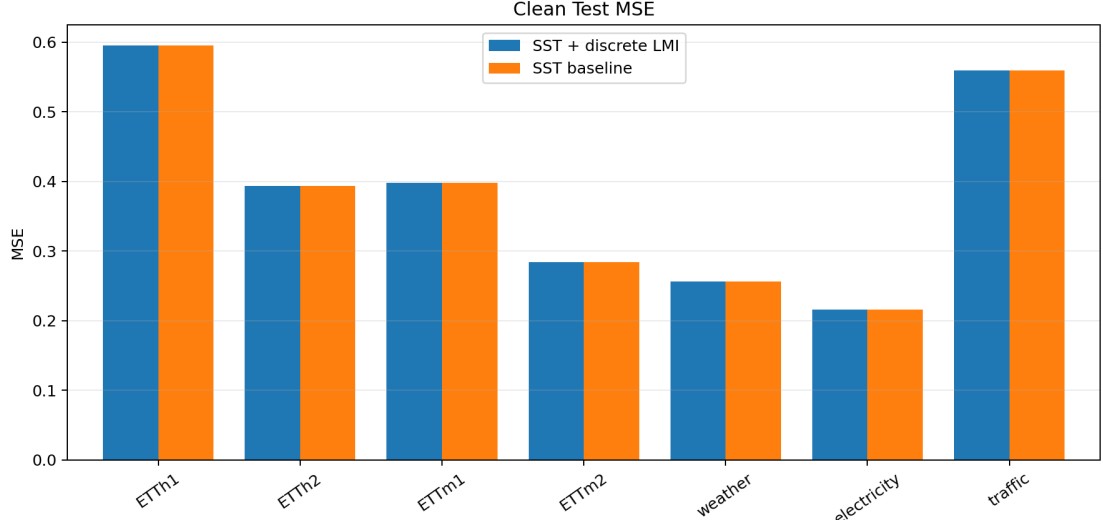

Figure 5: Per-run clean-test MSE delta vs. baseline of the LMI-regularized SST, sorted by dataset and horizon. Across 28/28 paired runs, the MSE delta is below 0.018%, with most deltas under 0.005%. The largest delta appears at `traffic`, pred_len = 720, the regime where the regularizer pushes hardest on the Mamba core.

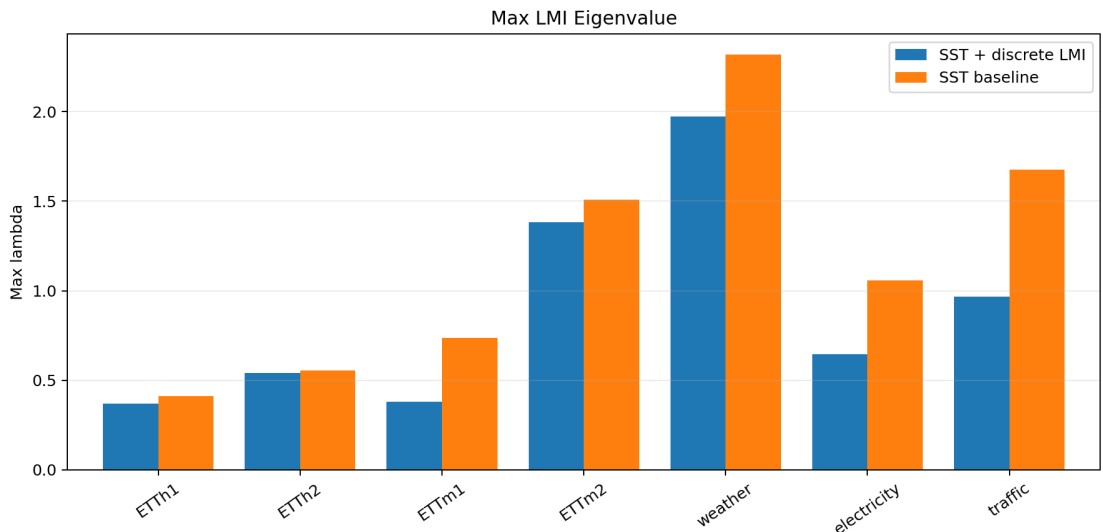

Figure 6: Per-dataset reduction in $\lambda_{\max}(\mathcal{L}_k)$ between baseline SST and the LMI-regularized variant, averaged across the four prediction horizons. The reduction is strict on every dataset; the residual positive value of $\lambda_{\max}$ is the basis for the "substantial reduction in violation, not certified passivity" framing in Section 8.6.

### C.3 Mild LMI-Weight Regime

The conservativeness ablation in Section 8.3 reported the strong LMI-weight regime $\lambda_{\mathrm{LMI}} = 10^{-2}$. Here we report the same comparison with the milder weight $\lambda_{\mathrm{LMI}} = 10^{-3}$, which probes how much of the conservative-

ness story depends on the strength of the regularizer. Aggregate results across the same 28 dataset–horizon pairs are summarized in Table 8.

| Variant | Clean MSE | Mean LMI viol. | $q_{\text{mean}}$ | $q_{\text{log-std}}$ |
|---|---|---|---|---|
| SST baseline | 0.385804 | 0.035824 | – | – |
| SST + discrete LMI ($P = I$, mild) | 0.385809 | 0.007541 | – | – |
| SST + discrete LMI (diag-$Q$, mild) | 0.385809 | 0.007537 | 1.0002 | $3.8 \times 10^{-4}$ |

Table 8: Mild-weight ($\lambda_{\text{LMI}} = 10^{-3}$) conservativeness ablation. The regularizer reduces mean Mamba-core LMI violation by approximately 79% in this regime, and diag-$Q$ remains a strict (if smaller) Pareto improvement over $P = I$ (28/28). The learned $q$ stays much closer to identity than under the strong weight, a useful sanity check that the diagonal mechanism is responsive to the regularizer strength.

The conservativeness conclusion of Section 8.3 carries through. Under the milder regime, the relative gap between $P = I$ and diag-$Q$ is smaller (paired-run violation reductions average $-0.05\%$ vs. the $-0.22\%$ of the strong regime) and the learned $q$ moves visibly less from identity. This is the expected sensitivity to the regularizer strength: when the LMI penalty is small, the optimizer does not need diagonal storage flexibility to satisfy it, and accordingly $q$ stays close to its initialization.

### C.4 Robustness Diagnostics: Per-Mode Breakdown

The aggregate robustness result of Section 8.4 averages across five injected perturbation modes. The corresponding state-norm picture is in Figure 7. The robust MSE picture, with all six variants superimposed, is in Figure 8.

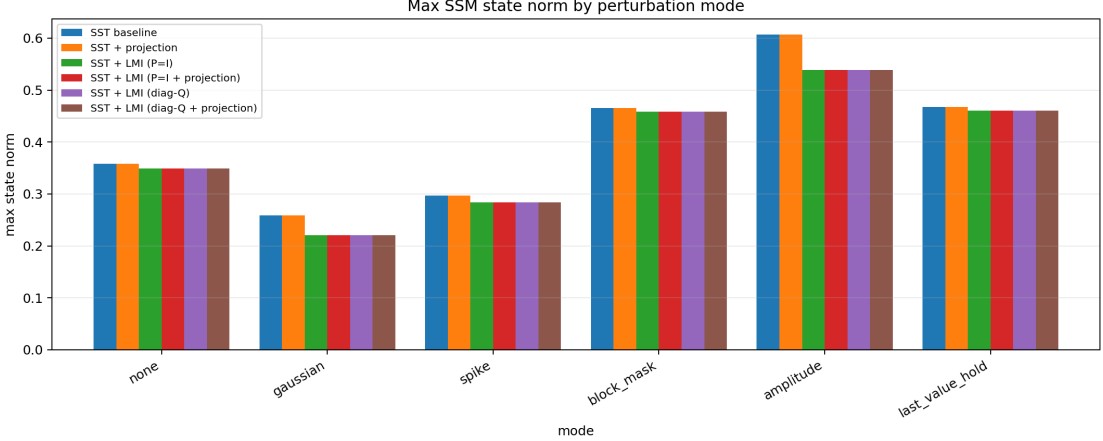

Figure 7: Mean max SSM state norm per dataset and per perturbation mode. The state-norm reduction is small in absolute terms (mean $-3.4\%$, peaking at $-11\%$ on the worst-state-norm runs) but holds in 35/35 pairs, providing a complementary internal diagnostic to the LMI-violation picture.

**On the projection variant.** The projection-only variant (no LMI regularizer, just the post-update projection) is statistically indistinguishable from the unregularized baseline across all robustness diagnostics in our runs. The LMI-with-projection variants (both $P = I$ and diag-$Q$) are statistically indistinguishable from their non-projection counterparts. We interpret this as evidence that the projection step is not binding in our setting and accordingly do not promote it as a meaningful design choice.

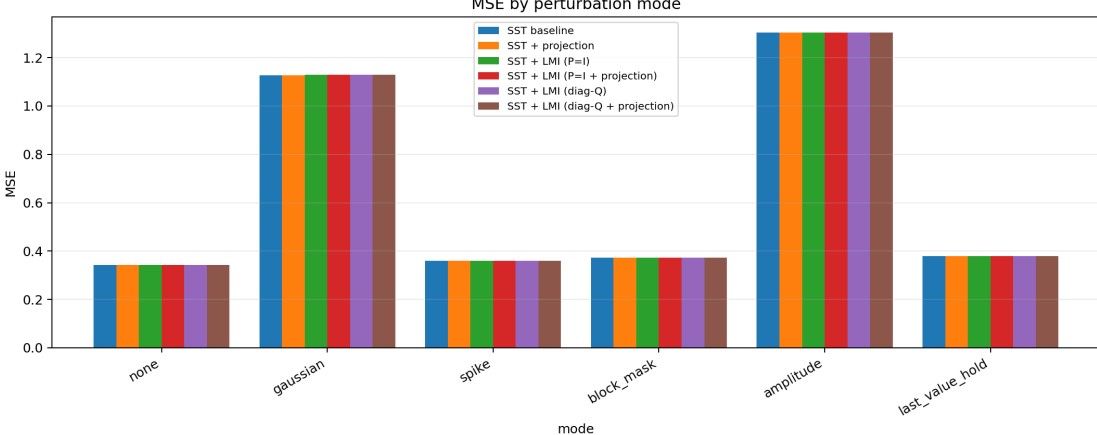

Figure 8: Robust forecasting MSE per dataset and per perturbation mode. All six variants (baseline, projection-only, $P = I$ LMI, $P = I$ + projection, diag-$Q$, diag-$Q$ + projection) overlap visibly on every cell. This is the visual face of the "no robust-MSE improvement" finding discussed in Section 8.4: the regularizer does not change end-to-end task error under perturbation, even though it changes Mamba-core internal diagnostics consistently.

### C.5 Router-Stability Ablation: Per-Dataset Detail

The aggregate result of Section 8.5 is that adding the router-stability penalty preserves clean MSE in 28/28 runs and consistently nudges the Mamba branch downward in 28/28 runs, but the train-time correlation between Mamba weight and LMI violation is essentially zero on aggregate (+0.0043). The per-dataset breakdown of this train-time correlation is in Figure 9; per-run deltas in $m_{\text{weight}}$ vs. the LMI-only variant are in Figure 10.

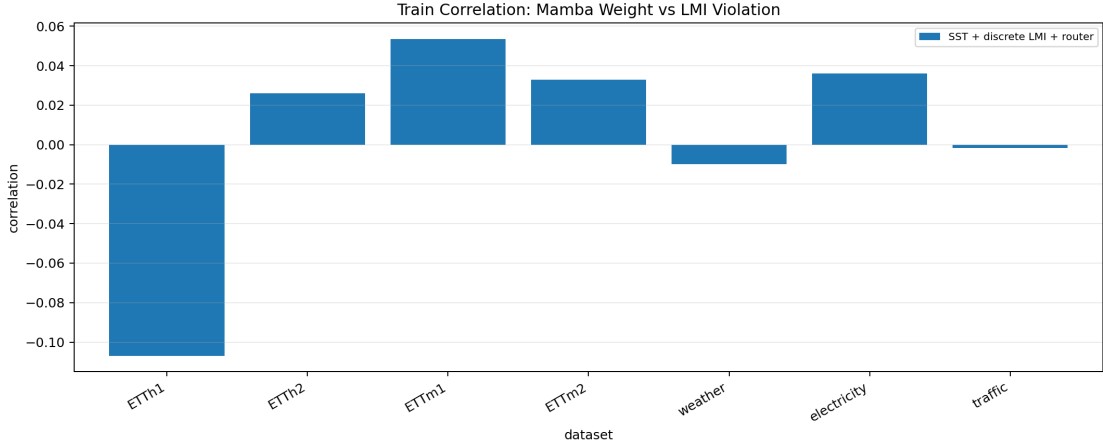

Figure 9: Per-dataset train-time correlation between the router's Mamba weight and the local LMI violation, under the energy-aware router penalty. The signs are inconsistent across datasets, ranging from $-0.107$ on ETTh1 (negative correlation, the desired energy-aware direction) to $+0.054$ on ETTm1 (positive correlation, the wrong direction). This dataset-level inconsistency is the precise reason we do not interpret the aggregate (+0.0043) as evidence of a learned energy-aware schedule.

**What the per-dataset breakdown adds.** On ETTh1 specifically, the train-time correlation is negative ($-0.107$) and consistent across all four horizons of that dataset. This shows the energy-aware mechanism

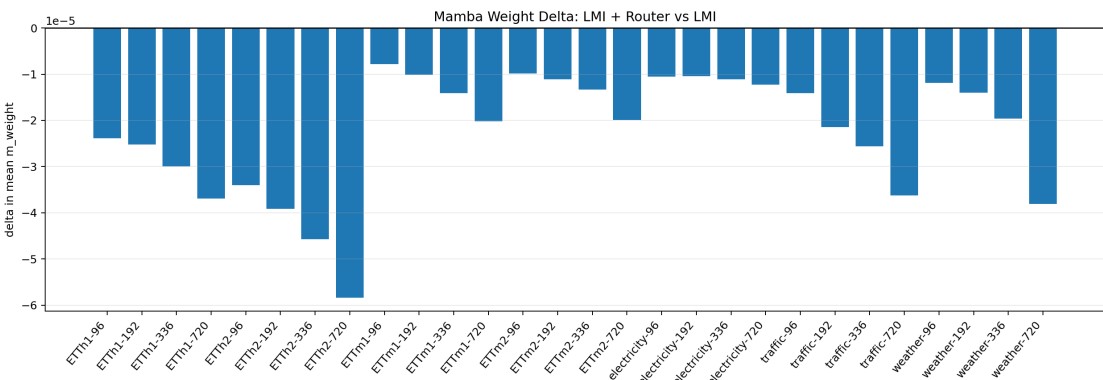

Figure 10: Per-run $m_{\text{weight}}$ delta of the router-energy variant relative to the LMI-only variant. Every bar is negative (28/28 runs nudge the Mamba branch downward), confirming the directional finding of Section 8.5, but the magnitudes are uniformly in the $10^{-5}$ to $10^{-4}$ range.

is in principle capable of producing the desired sign of correlation, but it does not generalize to a uniform negative-sign pattern across the other six datasets. We expect that the underlying issue is that the router is trained on clean inputs, where local LMI violations are not strongly correlated with task loss, so the router has no incentive (in the task-loss gradient) to actively avoid the Mamba branch when the Mamba core is locally near a violation. The most direct path to a genuinely learned, energy-aware schedule, as discussed in Section 9, is to retrain with injected perturbations so that violation magnitude becomes correlated with task loss; we leave this as future work.