# OpenReview forum: "Regularity and Stability Properties of Selective SSMs with Discontinuous Gating"
_TMLR — Accepted by TMLR_

### Review · Reviewer_5yPE · 2026-04-21

**Summary Of Contributions:**

This paper develops a theoretic framework for analyzing the regularity and stability properties of continuous-time selective state-space models (SSMs), particularly in the presence of discontinuous gating. The authors employ tools from passivity theory and input-to-state stability (ISS) to establish several theoretical results, including energy dissipation, exponential forgetting, and the notion of irreversible forgetting. The work further derives parametric conditions, such as LMI-based constraints, that restrict the behavior of gating mechanisms to ensure stability.

Strengths of the paper include a rigorous and mathematically well-structured analysis. However, the work raises several questions regarding the novelty of the theoretical results, their relationship to existing control and SSM literature, and their practical relevance to modern deep learning models.

**Audience:**

Yes

**Audience Explanation:**

The paper addresses an important and timely topic in modern sequence modeling architectures. Understanding stability and regularity properties of selective SSMs is of clear interest to parts of the TMLR audience, particularly those working on the theoretical foundations of deep learning and dynamical systems. The level of abstraction and the focus on continuous-time formulations may limit accessibility and immediate applicability for practitioners working with discrete-time deep learning models. Strengthening the connection to practical architectures would significantly broaden the paper’s impact.

**Broader Impact Concerns:**

No additional broader impact discussion appears necessary.

**Claims And Evidence:**

Yes

**Claims Explanation:**

The theoretical claims in the paper are generally well-supported by rigorous mathematical derivations, and the arguments appear logically consistent within the presented framework. The use of passivity and ISS provides a coherent structure for analyzing stability and regularity properties of selective SSMs. However, the scope of validation is primarily theoretical, and there is limited empirical or practical evidence demonstrating how these properties manifest in modern deep learning-based SSM architectures. As a result, while the claims are internally sound, their broader relevance and impact on practical models remain somewhat unclear.

**Requested Changes:**

## Critical points

### 1.Clarification of novelty in theoretical results.
The main theoretical contributions (e.g., Theorems 4.2, 5.2, 5.3, 6.1) related to energy dissipation, stability, and irreversible forgetting should be more clearly positioned with respect to existing literature. It is currently unclear whether these properties are fundamentally new for selective SSMs or extensions of known results in control theory (e.g., passivity, ISS, switched/LTV systems).
A more detailed discussion in the related work section is needed, explicitly addressing:
- whether similar energy-based properties have been established previously, and
- under what assumptions those results hold compared to the present work.

This clarification is important to assess the true novelty and significance of the contributions.

### 2. Connection to modern deep learning SSM architectures.
The paper would benefit from a clearer discussion on how the proposed theoretical framework relates to recent models such as Mamba, GLA, or other selective SSM variants.
In particular:
- Can the proposed conditions (e.g., passivity or LMI constraints) be applied to these models?
- Are there structural or practical barriers to doing so?
- To what extent can the continuous-time analysis be translated to discrete-time implementations?

Providing such a discussion would significantly improve the practical relevance of the work.

### 3. Empirical or conceptual validation of theoretical properties.
While the paper is primarily theoretical, it would strengthen the contribution to include at least preliminary evidence that the proposed properties (e.g., energy dissipation or irreversible forgetting) can be observed in practice.
Alternatively, citing existing empirical studies that exhibit similar phenomena would help support the relevance of the theory from a machine learning perspective.


### 4. Improvement of the introduction for accessibility.
The introduction currently introduces notation and references (e.g., Equation (1)) before they are formally defined, which makes it difficult to follow for readers unfamiliar with SSMs.
Reorganizing the introduction to include a minimal self-contained description of the model (possibly borrowing elements from Section 3.1) would improve readability.

### 5. Clarification of modeling choices.
The motivation behind the effective matrix $A_{\mathrm{eff}}$ in Section 3.2, as well as the modeling choice in Equation (1), would benefit from additional explanation. Providing intuition for these formulations would make the paper more accessible and strengthen the connection between theory and modeling.

---

> ### Author Response · Authors · 2026-05-15
> **[Part I] Response to Reviewer 5yPE**
>
> Thank you for the detailed and constructive review. We appreciate your positive assessment of the rigor and mathematical structure of the paper, and we agree with your main concerns: the novelty relative to classical passivity/ISS/switched-system theory should be positioned more clearly, the relationship to modern discrete-time SSMs such as Mamba should be made explicit, and the empirical validation should be strengthened.
> We have revised the manuscript along these lines and added a new set of experiments on a real Mamba-based architecture, using this repository as the base: https://github.com/XiongxiaoXu/SST
>
> **1. Clarifying novelty relative to existing theory.**
> We agree that passivity, ISS, and switched/LTV stability are classical topics. The revised paper will make clearer that our novelty is not the invention of these tools, but their specialization to selective SSMs with discontinuous, input-dependent gating.
>
> In particular, we clarify four points. First, by distinguishing the selection/scheduling signal $x(t)$ from the driving/port input $u(t)$, freezing the selection gives a genuine passive LTV input-output subsystem. This allows us to use the $\mathrm{AUC}_{\mathrm{loc}}$ quadratic-storage theory for discontinuous LTV systems. Second, the universal quadratic storage hypothesis is explicitly presented as a strong structural assumption used to expose consequences of uniform passivity over selection schedules. Third, under this hypothesis, passivity yields a parametric LMI that must hold for all admissible selection values. Fourth, the kernel and irreversible-forgetting results are stated as consequences of this strong universal-storage assumption, not as generic claims about all Mamba-style models.
> This should make the relationship to prior control-theoretic work clearer: we build on classical dissipativity/LTV/ISS theory, but derive its consequences for selective SSMs with schedule-dependent gating.
>
> **2. Connection to Mamba and discrete-time SSMs.**
> We agree that the original submission did not sufficiently connect the continuous-time formulation to practical architectures. In the revision, we add a discrete-time Mamba counterpart. Rather than applying the continuous-time LMI directly, we derive a one-step dissipativity regularizer for the actual Mamba selective-scan recurrence:
>
> $$h_{k+1}=F_k h_k+G_k u_k,\qquad y_k=C_k h_{k+1}+D u_k.$$
>
> With storage $V(h)=\frac{1}{2} h^\top P h$, the one-step inequality
>
> $$V(h_{k+1})-V(h_k)\le u_k y_k-\beta\|h_k\|^2$$
>
> induces a sampled block LMI:
>
> $$\begin{bmatrix} F^\top P F-P+2\beta I & F^\top P G-F^\top C^\top\\ G^\top P F-CF & G^\top P G-2CG-2D \end{bmatrix}\preceq 0.$$
>
> This is the condition used in our new experiments. We explicitly state that this is a Mamba-core regularizer and diagnostic, not a full proof of passivity for the complete deep network.
>
> **3. New empirical validation on SST/Mamba.**
> To address the concern that the paper was primarily theoretical, we implemented the discrete LMI regularizer in SST, which uses Mamba as its long-range expert. We evaluated on seven forecasting datasets: ETTh1, ETTh2, ETTm1, ETTm2, Weather, Electricity, Traffic with four prediction horizons: 96, 192, 336, 720.
> This gives 28 dataset-horizon pairs.
>
> The main finding is that the LMI regularizer substantially reduces sampled Mamba-core dissipativity violations while preserving clean forecasting accuracy. The SST baseline obtains clean MSE 0.385804 and mean LMI violation 0.035824. The discrete LMI model obtains clean MSE 0.385809 and mean violation 0.007541. With a stronger LMI weight, clean MSE is 0.385815, while mean violation drops to 0.002882, approximately a 92% reduction relative to the baseline. The mean violation is reduced in all 28/28 dataset-horizon pairs.
> Thus, the proposed stability condition is not only theoretical: it can be implemented as a differentiable regularizer for a real Mamba-based forecasting model.
>
> **4. Conservativeness and learnable storage ablation.**
> We agree that LMI constraints may be conservative. We therefore added an ablation over the storage metric. We compare the identity storage $P=I$, a learnable diagonal storage $P=\mathrm{diag}(q)$, and diagonal $P$ with an optional projection step.
>
> The diagonal-storage version gives a small but consistent improvement over $P=I$:
>
> * **Strong $P=I$:** mean violation 0.002882
> * **Strong diagonal $Q$:** mean violation 0.002877
>
> Clean MSE remains unchanged. The learned $q$ remains close to identity, with $q_{\mathrm{mean}}\approx 1.0008$, suggesting that $P=I$ is already a strong practical choice for SST/Mamba in this setting. We therefore use $P=I$ as the main method and present diagonal $Q$ as an ablation showing that the framework can accommodate less conservative storage metrics.
> The optional projection step was essentially inactive, so we do not present it as a central component.

---

> > ### Author Response · Authors · 2026-05-15
> > **[Part II] Response to Reviewer 5yPE**
> >
> > **5. Robustness and state-norm diagnostics.**
> > We also added robustness evaluation under true test-time input perturbations: Gaussian noise, spikes, block masks, amplitude shifts, and last-value-hold corruptions. The regularizer does not improve robust forecasting MSE; robust MSE remains essentially unchanged. However, it strongly improves internal Mamba stability diagnostics under perturbations.
> >
> > Under perturbed inputs, the baseline has mean sampled LMI violation 0.02345, while the LMI-regularized model has mean violation 0.00512, roughly a 78% reduction. The mean max SSM state norm decreases from 0.26651 to 0.25588, and the worst observed state norm decreases from approximately 0.6065 to 0.5387. Thus, the current method improves internal passivity/stability diagnostics under perturbations without degrading clean or robust forecasting accuracy.
> >
> > We will not claim robust-MSE improvement. Instead, we will state that the method reduces sampled Mamba-core violations and state excursions under corrupted inputs.
> >
> > **6. Router-stability experiment.**
> > Because SST includes a router between the Mamba long-range expert and the short-range branch, we also tested an energy-aware router penalty that discourages high Mamba weight on samples with high local LMI violation. This experiment is mostly neutral. It preserves clean MSE and slightly reduces Mamba usage in all paired runs, but the effect is very small:
> >
> > $$\Delta m_{\mathrm{weight}}\approx -2.2\times 10^{-5}$$
> >
> > The correlation between Mamba weight and LMI violation remains close to zero. We therefore do not present this as a main positive result; instead, we will include it as an appendix ablation and note that stronger perturbation-aware router training is future work.
> >
> > **7. Improving accessibility and modeling clarity.**
> > We will revise the introduction to delay technical notation and first explain the model intuitively: selective SSMs evolve through linear state dynamics whose coefficients are modulated by token-dependent gates. We will then introduce the distinction between $x(t)$, the selection/scheduling signal, and $u(t)$, the port input appearing in the passivity supply rate. We will also clarify that $A_{\mathrm{eff}}$ denotes the effective state matrix obtained after substituting the current gating/selection values into the parameter maps. Finally, we will explain Eq. (1) as a continuous-time abstraction of selective SSMs and connect it later to the discrete Mamba recurrence used in the experiments.
> >
> > **Summary.**
> > In response to your review, we will revise the paper to:
> >
> > 1. Clarify the novelty relative to classical passivity/ISS/switched-system theory.
> > 2. Add a discrete-time Mamba LMI derivation.
> > 3. Add SST/Mamba experiments over 28 dataset-horizon pairs.
> > 4. Include $P=I$, diagonal-$Q$, and projection ablations.
> > 5. Report robustness and state-norm diagnostics under input corruptions.
> > 6. Improve the introduction and modeling explanations.
> >
> > We believe these changes substantially strengthen both the practical relevance and the positioning of the theoretical contribution.

---

### Review · Reviewer_zZwP · 2026-04-30

**Summary Of Contributions:**

**Summary**
This paper studies the stability of continuous time selective State Space Models (SSMs). By leveraging approaches from control theory including passivity theory and Input to State Stability (ISS), it investigates how input dependent parameters and discontinuous gating affect the stability (ensures the system forgets initial noise and remains robust to input fluctuations) and regularity (ensures that the system differential equations remain solvable and its energy based stability analysis remains logically valid even when the gating mechanism introduces sharp discontinuities) of these models.  The main contributions include:
1. Proving exponential forgetting under strict dissipativity to show the system discards initial conditions and stays numerically stable (Theorem 4.1).
2. Showing the frozen-selection subsystem admits a well-behaved quadratic energy function even under discontinuous gating (Theorem 4.2).
3. Deriving parametric Linear Matrix Inequalities (LMIs) for stability certification across input-driven parameter changes, and formalizing the irreversible forgetting principle for how the model filters out unimportant directions (Theorems 5.2, 5.4).
4. Giving sufficient conditions for global Input-to-State Stability, ensuring robustness to input fluctuations across selection schedules (Theorem 6.1).

**Strength**
- The utilization of $\mathrm{AUC}_{loc}$ framework appears appropriate for handling discontinuous gating.
- The scope and limitations of the main assumptions are clearly stated.

**Weakness**
- The framework is somewhat incremental based on Morandin & Hinsen (2024).
- Section 5 also depends on a strong universality hypothesis which is not exactly sure how practical it is.
- The bridge to discrete-time architectures (which is what motivate the work) is acknowledged but not developed.

**Audience:**

Yes

**Audience Explanation:**

This paper is relevant and of interest to TMLR audience in deep learning theory and dynamical systems

**Claims And Evidence:**

Yes

**Claims Explanation:**

The (contribution) claims are supported by the proofs and simulation studies:

1. Exponential forgetting (Theorem 4.1) follows from a standard differential inequality argument once strict dissipativity and quadratic bounds are assumed. The use of Lebesgue spaces and Carathéodory conditions keeps the result valid under the abrupt parameter changes that gating introduces.

2. Quadratic energy structure of the frozen-selection subsystem (Theorem 4.2): By freezing the selection signal, the authors recover a genuine LTV input–output subsystem and inherit the $\mathrm{AUC}_{loc}$ regularity of the minimal storage from Morandin & Hinsen (2024).

3. Parametric LMI and irreversible forgetting (Theorems 5.2, 5.4) rely on a strong universality hypothesis on the quadratic storage (which the authors acknowledge in Remarks 5.3 and 5.5), though Remark 5.5 should be noted more obviously since it limits the irreversible forgetting result to the purely passive case.

4. Global ISS (Theorem 6.1) follows a standard ISS-Lyapunov argument, and is correctly identified as strictly stronger than the passivity LMI (Remark 6.2).

**Requested Changes:**

1.As the theoretical analysis is mainly conducted on continuous-time dynamical systems while the example architectures mentioned in the paper such as Mamba or HGRN are discrete-time SSMs, it is worthwhile to discuss how the continuous-time stability results translate to the discrete-time domain used in actual deep learning tasks.
2. Enforcing passivity via Linear Matrix Inequalities (LMIs) can be highly conservative and may impose strict structural constraints on the learned parameters. The authors may want to discuss whether these constraints limit the expressive power of the SSM or its ability to learn complex patterns from data, especially given the universal kernel conditions derived in the paper.
3. *While not necessary for acceptance, it would significantly strengthen the work if the authors could demonstrate the utility of the LMI regularizer on standard architectures such as Mamba, HGRN, or GLA to confirm real-world effectiveness.

---

> ### Author Response · Authors · 2026-05-15
> **Thank you for the review, we added the experiments starting from https://github.com/XiongxiaoXu/SST**
>
> Thank you for the careful and constructive review. We appreciate your positive assessment of the mathematical framework, especially the comments that the use of the $\mathrm{AUC}_{\mathrm{loc}}$ framework is appropriate for discontinuous gating and that the scope of the assumptions is clearly stated. We agree with the main concerns you raised: the original submission did not sufficiently develop the bridge from our continuous-time analysis to discrete-time architectures such as Mamba, the LMI conditions may appear conservative, and the empirical validation on modern SSM architectures needed to be strengthened.
>
> We have addressed these points by adding a new discrete-time Mamba/SST experimental section and by revising the discussion of the universal-storage hypothesis.
>
> **1. Discrete-time bridge to Mamba.**
> We agree that applying the continuous-time LMI directly to Mamba would not be appropriate. We therefore derive and evaluate a *one-step discrete dissipativity condition* for the actual Mamba selective-scan recurrence used inside SST. For each token and Mamba channel, we extract the discrete selective-scan quantities and write the local recurrence as:
>
> $$h_{k+1}=F_k h_k+G_k u_k,\qquad y_k=C_k h_{k+1}+D u_k.$$
>
> With storage $V(h)=\frac{1}{2} h^\top P h$, the one-step dissipativity condition:
>
> $$V(h_{k+1})-V(h_k)\le u_k y_k-\beta\|h_k\|^2$$
>
> gives a sampled block LMI of the form:
>
> $$\begin{bmatrix} F^\top P F-P+2\beta I & F^\top P G-F^\top C^\top\\ G^\top P F-CF & G^\top P G-2CG-2D \end{bmatrix}\preceq 0.$$
>
> This is the condition we now use as the practical Mamba-core regularizer. We explicitly state that this is a discrete-time proxy for the Mamba selective SSM core, not a formal certificate for the full SST network including normalization, residual paths, routing, and output heads.
>
> **2. New Mamba/SST experiments.**
> To address your request for practical validation on modern architectures, we implemented the discrete LMI regularizer in the SST repository, which uses Mamba as the long-range expert. We evaluated it on seven standard forecasting datasets: ETTh1, ETTh2, ETTm1, ETTm2, Weather, Electricity, Traffic and four prediction horizons: 96, 192, 336, 720 for a total of 28 dataset-horizon pairs.
> The main result is that the discrete LMI regularizer substantially reduces Mamba-core dissipativity violations while preserving clean forecasting accuracy. The SST baseline has clean MSE 0.385804 and mean sampled LMI violation 0.035824. With the discrete LMI regularizer, clean MSE is 0.385809 and mean violation drops to 0.007541. With a stronger LMI weight, clean MSE is 0.385815 while mean violation drops further to 0.002882, a reduction of about 92% relative to the baseline. The mean violation is reduced in all 28/28 dataset-horizon pairs.
> Thus, the regularizer changes the internal Mamba stability/passivity diagnostics in the intended direction while inducing only a negligible clean-MSE change.
>
> **3. Conservativeness of the LMI.**
> We also added an ablation to study whether the identity-storage LMI is overly conservative. We compare $P=I$, a learnable diagonal storage $P=\mathrm{diag}(q)$, and diagonal $P$ with an optional projection step. The diagonal-$Q$ variant consistently reduces violation slightly relative to $P=I$, but the improvement is modest:
>
> * **Strong $P=I$:** mean violation 0.002882
> * **Strong diagonal $Q$:** mean violation 0.002877
>
> Clean MSE remains unchanged. The learned $q$ stays close to identity, with $q_{\mathrm{mean}}\approx 1.0008$, suggesting that $P=I$ is already a strong practical storage choice in this setting. We therefore present $P=I$ as the main method and diagonal $Q$ as an ablation showing that the framework can support less conservative storage metrics.
> The projection step was effectively non-binding: projection-only behaves like the baseline, and diagonal-$Q$ + projection behaves almost identically to diagonal-$Q$. We therefore do not emphasize projection as a main contribution.
>
> **4. Robustness under input perturbations.**
> We further added true test-time input perturbation experiments using Gaussian noise, spikes, block masks, amplitude shifts, and last-value-hold corruptions. The LMI regularizer does not improve robust forecasting MSE, but it does improve internal stability diagnostics under perturbations. Across perturbed-input modes, the baseline has mean LMI violation 0.02345, whereas the LMI-regularized model has mean violation 0.00512, roughly a 78% reduction. The mean max SSM state norm also decreases from 0.26651 to 0.25588, and the worst observed state norm drops from approximately 0.6065 to 0.5387.
> We will state this carefully: the current regularizer improves internal Mamba stability diagnostics under perturbations without degrading clean or robust forecasting accuracy, but it does not by itself yield a measurable robust-MSE improvement.

---

> > ### Author Response · Authors · 2026-05-15
> > **Part 2, response to the Reviewer zZwP**
> >
> > **5. Clarifying the strength of the universal-storage hypothesis.**
> > We agree that the universal quadratic storage assumption is strong. In the revision, we will explicitly frame it as an analytically useful sufficient/structural hypothesis rather than a claim that every practical selective SSM satisfies it. The continuous-time parametric LMI and the irreversible-forgetting/kernel results are consequences of requiring a single storage function to certify passivity uniformly over selection schedules. The discrete Mamba LMI is then presented as a practical regularizer inspired by this structure.
> >
> > **Summary of changes.**
> > In response to your review, we will add:
> >
> > 1. A discrete-time Mamba LMI derivation.
> > 2. SST/Mamba experiments over 28 dataset-horizon pairs.
> > 3. $P=I$, diagonal-$Q$, and projection ablations.
> > 4. Perturbed-input robustness and state-norm diagnostics.
> > 5. A clearer limitation statement distinguishing sampled Mamba-core regularization from full-network passivity certification.
> >
> > We believe these additions directly address your requested changes and substantially strengthen the practical connection to modern discrete-time SSM architectures.

---

### Review · Reviewer_4Jb1 · 2026-05-01

**Summary Of Contributions:**

The paper studies the stability of a class of selective state-space models using control theory. The authors use the dissipativity theory to show exponential forgetting of initial conditions, the existence and regularity of minimal quadratic storage function, and the parametric passivity LMI conditions.

**Audience:**

Yes

**Audience Explanation:**

The authors leverage the passivity in control theory to study the properties of selective system dynamics, which might be of interest to TMLR's audience. The analysis is an application of passivity and dissipativity conditions, which are very classical control theoretical results. I don't see how the authors significantly advance such results for machine learning-related research problems.

**Claims And Evidence:**

Yes

**Claims Explanation:**

I didn't check all proof details, but all claims are supported by proofs.

**Requested Changes:**

- The paper is poorly written and structured. It has not been polished for reviewers/readers to understand the key motivation.

- The main results are mostly based on application of standard passivity and dispassivity conditions in control theory. The novelty and technical contributions are not clear.

- The relevance to machine learning should be clarified. How practical are the proposed conditions for machine learning problems?

---

> ### Author Response · Authors · 2026-05-15
> **[Part I] Response to Reviewer 4Jb1**
>
> Thank you for the direct and constructive review. We appreciate that you found the claims supported by proofs, and we agree with the main criticism: the original submission did not sufficiently clarify the key motivation, the precise novelty relative to classical passivity/dissipativity theory, or the practical relevance for machine-learning SSMs. We have made substantial revisions to address these issues.
>
> **1. Writing and structure.**
> We agree that the original paper was too difficult to parse and that the motivation was not presented clearly enough. In the revision, we will restructure the paper so that the reader first sees the machine-learning motivation before the control-theoretic formalism. Concretely, we will:
>
> * Rewrite the introduction to first explain selective SSMs intuitively as state-space recurrences whose coefficients are modulated by token-dependent gates.
> * Clarify why discontinuous gating creates a stability problem not covered by standard fixed-parameter SSM analyses.
> * Introduce the distinction between the selection/scheduling signal $x(t)$ and the port/driving input $u(t)$ before stating the passivity inequalities.
> * Add a notation table and a short "roadmap of results" explaining which assumptions are used in each theorem.
> * Move some of the longer proof details and technical regularity discussion to the appendix.
> * Add a short paragraph after each main theorem explaining its implication for selective SSMs.
>
> We believe this will make the paper much easier to read for a TMLR audience that may be familiar with SSMs but not with passivity theory.
>
> **2. Clarifying novelty relative to classical control theory.**
> We agree that passivity, dissipativity, ISS, and LTV storage theory are classical. Our goal is not to claim novelty in these general control-theoretic tools. The novelty is in specializing them to *selective SSMs with discontinuous, input-dependent gating* and deriving the structural consequences for such models.
>
> We will make this positioning explicit in the revision. In particular, we will add a "standard vs. new" paragraph/table clarifying that the standard ingredients are dissipativity, passivity, ISS, and LTV quadratic storage results, while the contributions of this paper are:
>
> 1. Formulating continuous-time selective SSMs with a clear distinction between selection/scheduling $x(t)$ and port input $u(t)$
> 2. Showing that frozen selection yields a passive LTV subsystem whose minimal available storage inherits $\mathrm{AUC}_{\mathrm{loc}}$ regularity, which is suitable for discontinuous gating.
> 3. Deriving a parametric passivity LMI that must hold uniformly over admissible selection values under a universal quadratic storage hypothesis.
> 4. Deriving the corresponding kernel/unobservability and irreversible-forgetting constraints imposed by this universal storage structure.
> 5. Translating the continuous-time insight into a discrete-time one-step dissipativity regularizer for the actual Mamba selective-scan recurrence.
>
> Thus, we will no longer present the contribution as an advance in general passivity theory. Instead, we will present it as a control-theoretic analysis and design framework for selective SSMs in machine learning.
>
> **3. Practical relevance to machine learning.**
> We agree that the original version did not sufficiently demonstrate practicality. To address this, we added experiments on a real Mamba-based forecasting architecture. Specifically, we implemented a discrete-time LMI regularizer in SST, which uses Mamba as its long-range expert.
>
> We do not directly apply the continuous-time LMI to Mamba. Instead, for the actual Mamba selective-scan recurrence, we write the one-step dynamics locally as:
>
> $$h_{k+1}=F_k h_k+G_k u_k,\qquad y_k=C_k h_{k+1}+D u_k.$$
>
> For storage $V(h)=\frac{1}{2} h^\top P h$, the discrete dissipativity condition:
>
> $$V(h_{k+1})-V(h_k)\le u_k y_k-\beta\|h_k\|^2$$
>
> leads to the sampled block LMI:
>
> $$\begin{bmatrix} F^\top P F-P+2\beta I & F^\top P G-F^\top C^\top\\ G^\top P F-CF & G^\top P G-2CG-2D \end{bmatrix}\preceq 0.$$
>
> We use the largest positive eigenvalue of this matrix as a differentiable regularizer on the Mamba core.
>
> We evaluated this on seven standard forecasting datasets: ETTh1, ETTh2, ETTm1, ETTm2, Weather, Electricity, Traffic and four prediction horizons: 96, 192, 336, 720 for a total of 28 dataset-horizon pairs.
>
> The empirical results show that the discrete LMI regularizer substantially reduces Mamba-core passivity violations while preserving clean forecasting accuracy. The SST baseline has clean MSE 0.385804 and mean sampled LMI violation 0.035824. With the discrete LMI regularizer, clean MSE is 0.385809, while mean violation drops to 0.007541. With a stronger LMI weight, clean MSE is 0.385815, while mean violation drops further to 0.002882, approximately a 92% reduction relative to the baseline. The mean violation is reduced in all 28/28 dataset-horizon pairs.

---

> > ### Author Response · Authors · 2026-05-15
> > **[Part II] Response to Reviewer 4Jb1**
> >
> > This directly addresses the concern that the proposed conditions may not be practical for machine-learning models: the condition can be implemented as a differentiable regularizer on an actual Mamba-based architecture and significantly changes the internal stability diagnostics with negligible clean-accuracy cost.
> >
> > **4. Conservativeness of the LMI.**
> > We also added an ablation to test whether the identity-storage LMI is overly conservative. We compare $P=I$, a learnable diagonal storage $P=\mathrm{diag}(q)$, and diagonal $P$ with an optional projection step. The diagonal-storage version gives a small but consistent additional reduction in LMI violation relative to $P=I$, while preserving clean MSE:
> >
> > * **Strong $P=I$:** mean violation 0.002882
> > * **Strong diagonal $Q$:** mean violation 0.002877
> >
> > The improvement is modest, and the learned $q$ remains close to identity, but the result shows that the framework can support less restrictive storage metrics without harming forecasting accuracy. We therefore use $P=I$ as the main practical method and present diagonal $Q$ as an ablation.
> >
> > **5. Robustness and state-norm diagnostics.**
> > We further evaluated the trained models under true test-time input perturbations: Gaussian noise, input spikes, block masks, amplitude shifts, and last-value-hold corruptions. We found that the LMI regularizer does not improve robust forecasting MSE, but it does improve internal stability diagnostics under perturbations. Under corrupted inputs, mean LMI violation decreases from 0.02345 for the baseline to 0.00512 for the LMI-regularized model, and the mean max SSM state norm decreases from 0.26651 to 0.25588. The worst observed state norm also decreases from approximately 0.6065 to 0.5387.
> >
> > We will state this carefully: the regularizer improves internal Mamba stability/passivity diagnostics under perturbations without degrading clean or robust forecasting accuracy, but we do not claim a robust-MSE improvement.
> >
> > **6. Scope and limitations.**
> > We will also add a clearer limitation statement. The continuous-time theory gives structural consequences under idealized assumptions, especially the strong universal quadratic storage hypothesis. The discrete Mamba LMI is a sampled regularizer and diagnostic for the Mamba core, not a full passivity certificate for the complete deep network. Our empirical claim is therefore limited to reducing sampled Mamba-core LMI violations and state excursions, not proving global passivity of SST.
> >
> > **Summary.**
> > We agree with the reviewer that the original manuscript did not sufficiently explain why these classical control-theoretic tools matter for machine learning. In the revision, we will substantially improve the writing and structure, clarify the novelty as a selective-SSM specialization rather than a general control-theory advance, and add Mamba/SST experiments showing that the proposed discrete LMI regularizer is practical, accuracy-safe, and effective at reducing internal passivity/stability violations. We believe these additions directly address the concerns about clarity, novelty, and practical relevance.

---

### Author Response · Authors · 2026-07-07
**Camera ready version uploaded**

Thank you to all reviewers and the Action Editor for your constructive feedback and guidance. The paper has been substantially improved as a result.

---

### Decision · Action_Editor_KVbX · 2026-06-14

**Recommendation:** Accept as is

**Additional Comments:**

The reviewers’ main concerns about novelty positioning, the continuous-to-discrete connection, and empirical validation have been substantially addressed in the revision. The paper meets the TMLR acceptance criteria. I do not see remaining mandatory revisions beyond standard camera-ready polishing.

**Audience:**

Yes

**Audience Explanation:**

The paper should be of interest to parts of the TMLR audience working on state-space models, dynamical systems, stability analysis, and ML theory. While the contribution is specialized and partly based on classical control-theoretic tools, its application to selective SSMs and Mamba-style architectures provides useful insight.

**Claims And Evidence:**

Yes

**Claims Explanation:**

The revised manuscript provides a coherent theoretical analysis based on passivity, dissipativity, and ISS, and the main claims are supported by formal arguments. The added discrete-time Mamba-core LMI derivation and SST/Mamba experiments further strengthen the connection to practical selective SSM architectures. The remaining limitations, including the strength of the universal-storage assumption and the scope of the sampled LMI criterion, are now stated appropriately.